# BehaviorGPT: Smart Agent Simulation for Autonomous Driving with Next-Patch Prediction

**Zikang Zhou**[1*]     **Haibo Hu**[1*]     **Xinhong Chen**[1]     **Jianping Wang**[1]     **Nan Guan**[1]
**Kui Wu**[2]     **Yung-Hui Li**[3]     **Yu-Kai Huang**[4]     **Chun Jason Xue**[5]
[1]City University of Hong Kong     [2]University of Victoria     [3]Hon Hai Research Institute
[4]Carnegie Mellon University     [5]Mohamed bin Zayed University of Artificial Intelligence
{zikanzhou2-c, haibohu2-c}@my.cityu.edu.hk
{xinhong.chen, jianwang, nanguan}@cityu.edu.hk
wkui@uvic.ca   yunghui.li@foxconn.com   yukaih2@andrew.cmu.edu
jason.xue@mbzuai.ac.ae

## Abstract

Simulating realistic behaviors of traffic agents is pivotal for efficiently validating the safety of autonomous driving systems. Existing data-driven simulators primarily use an encoder-decoder architecture to encode the historical trajectories before decoding the future. However, the heterogeneity between encoders and decoders complicates the models, and the manual separation of historical and future trajectories leads to low data utilization. Given these limitations, we propose BehaviorGPT, a homogeneous and fully autoregressive Transformer designed to simulate the sequential behavior of multiple agents. Crucially, our approach discards the traditional separation between "history" and "future" by modeling each time step as the "current" one for motion generation, leading to a simpler, more parameter- and data-efficient agent simulator. We further introduce the Next-Patch Prediction Paradigm (NP3) to mitigate the negative effects of autoregressive modeling, in which models are trained to reason at the patch level of trajectories and capture long-range spatial-temporal interactions. Despite having merely 3M model parameters, BehaviorGPT won first place in the 2024 Waymo Open Sim Agents Challenge with a realism score of 0.7473 and a minADE score of 1.4147, demonstrating its exceptional performance in traffic agent simulation.

**Keywords:** Multi-Agent Systems, Transformers, Generative Models, Autonomous Driving

## 1   Introduction

Autonomous driving has emerged as an unstoppable trend, with its rapid development increasing the demand for faithful evaluation of autonomy systems' reliability [32]. While on-road testing can measure driving performance by allowing autonomous vehicles (AVs) to interact with the physical world directly, the high testing cost and the scarcity of safety-critical scenarios in the real world have hindered large-scale and comprehensive evaluation. As an alternative, validating system safety via simulation has become increasingly attractive [14, 48, 53, 44] as it enables rapid testing in diverse driving scenarios simulated at a low cost. This work focuses on smart agent simulation, i.e., simulating the behavior of traffic participants such as vehicles, pedestrians, and cyclists in the digital world, which is critical for efficiently validating and iterating behavioral policies for AVs.

A good simulator should be realistic, matching the real-world distribution of multi-agent behaviors to support the assessment of AVs' ability to coexist with humans safely. To this end, researchers started

---

*Equal contribution.

38th Conference on Neural Information Processing Systems (NeurIPS 2024).

by designing naive simulators that mainly replay the driving logs collected in the real world [27, 29]. When testing new driving policies that deviate from the ones during data collection, agents in such simulators often exhibit unrealistic interactions with AVs, owing to the lack of reactivity to AVs' behavior changes. To simulate reactive agents, traditional approaches [14, 28] apply traffic rules to control agents heuristically [45, 26], which may struggle to capture real-world complexity. Recently, the availability of large-scale driving data [6, 15, 50], the emergence of powerful deep learning tools [19, 47, 20], and the prosperity of related fields such as motion forecasting [16, 46, 59, 42, 58], have spurred the development of data-driven agent simulation [44, 4, 24, 52, 56] towards more precise matching of behavioral distribution. With the establishment of standard benchmarks like the Waymo Open Sim Agents Challenge (WOSAC) [32], which systematically evaluates the realism of agent simulation in terms of kinematics, map compliance, and multi-agent interaction, the research on data-driven simulation approaches has been further advanced [49, 35].

Existing learning-based agent simulators [44, 4, 24, 52, 56, 49, 35] mainly mirror the techniques from motion forecasting [16, 46, 59, 42, 58, 41, 18] and opt for an encoder-decoder architecture, presumably due to the similarity between the two fields. Typically, these models use an encoder to extract historical information and a decoder to predict agents' future states leveraging the encoded features. This paradigm requires manually splitting the multi-agent time series into a historical and a future segment, with the two segments being processed by separate encoders and decoders with heterogeneous architecture. For example, MVTA [49] constructs training samples by randomly selecting a "current" timestamp to divide sequences into historical and future components. Others [52, 35] use fixed-length agent trajectories as historical scene context, conditioned on which the multi-agent future is sampled from the decoder. Nonetheless, the benefit of employing heterogeneous modules to separately encode the history and decode the future, at the cost of significantly complicating the architecture, is unclear. Moreover, the manual separation of history and future leads to low utilization of data and computation: as every point in the sequence can be used for the separation, we believe a sample-efficient framework should be able to learn from every possible history-future pair from the sequence in parallel, which cannot be easily achieved by encoder-decoder solutions owing to their heterogeneous processing for the historical and the future time steps.

Inspired by the success of decoder-only Large Language Models (LLMs) [37, 38, 5], we introduce a fully autoregressive Transformer architecture, dubbed BehaviorGPT, into the field of smart agent simulation to overcome the limitations of previous works. By applying homogeneous Transformer blocks [47] to the complete trajectory snippets without differentiating history and future, we arrive at a simpler, more parameter-efficient, and more sample-efficient solution for agent simulation. Utilizing relative spacetime representations [58], BehaviorGPT symmetrically models each agent state in the sequence as if it were the "current" one and tasks each state with modeling subsequent states' distribution during training. As a result, our framework maximizes the utilization of traffic data for autoregressive modeling, avoiding wasting any learning signals available in the time series.

Autoregressive modeling with imitation learning, however, suffers from compounding errors [39] and causal confusion [11]. Concerning the behavior simulation task, we observed that blindly mimicking LLMs' training paradigm of next-token prediction [35], regardless of the difference in tokens' semantics across tasks, will make these issues more prominent. For a next-token prediction model embedding tokens at 10 Hz, a low training loss can be achieved by simply copying and pasting the current token as the next one without performing any long-range interaction reasoning in space or time. To mitigate this issue, we introduce the Next-Patch Prediction Paradigm (NP3) that enables models to reason at the patch level of trajectories, as illustrated in Figure 1. By enforcing models to autoregressively generate the next trajectory patch containing multiple time steps, which requires understanding the high-level semantics of agent behaviors and capturing long-range spatial-temporal interactions, we prevent models from leveraging trivial shortcuts during training. We equip BehaviorGPT with NP3 and attain superior performance on WOSAC [32] with merely 3M model parameters, demonstrating the effectiveness of our modeling framework for smart agent simulation.

Our main contributions are three-fold. First, we propose a fully autoregressive architecture for smart agent simulation, which consists of homogeneous Transformer blocks that process multi-agent long sequences with high parameter and sample efficiency. Second, we develop the Next-Patch Prediction scheme to enhance long-range interaction reasoning, leading to more realistic multi-agent simulation over a long horizon. Third, we achieve remarkable performance on the Waymo Open Motion Dataset, winning first place in the 2024 Waymo Open Sim Agents Challenge.

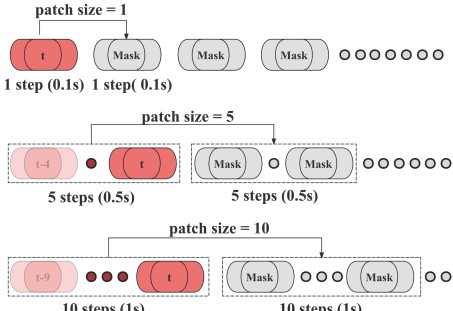

Figure 1: **Next-Patch Prediction Paradigm** with patch sizes of 1, 5, and 10 time steps for trajectories sampled at 10 Hz. The capsules in dark red represent the agent states at the current time step $t$, while the faded red capsules indicate agents' past states. The grey circles represent the masked agent states required for generation. Our approach groups multi-step agent states as patches, demanding each patch to predict the subsequent patch during training.

## 2  Related Work

### 2.1  Multi-Agent Traffic Simulation

Multi-agent traffic simulation is essential for developing and testing autonomous driving systems. From early systems like ALVINN [36] to contemporary simulators such as CARLA [14] and SUMO [28], these platforms have used heuristic driving policies to simulate agents' reactive behaviors [8, 7, 10]. However, they struggle to capture real-world complexity since policies based on simple heuristics are not robust enough to handle all sorts of scenarios. With the availability of large-scale data and deep learning approaches, generative models like VAEs [44], GANs [24], Diffusion [56], and autoregressive models [49, 41, 35] have gained success in generating multi-agent motions, which greatly enhance the realism of simulations. Given the temporal dependency of agent trajectories, autoregressive models naturally fit the simulation task, while others require extra designs to capture such dependencies. Among the existing autoregressive models, two representatives are MotionLM [41] and Trajeglish [35]. Both of them adopt an encoder-decoder paradigm, designing complicated scene context encoders to extract historical information before autoregressive decoding. In contrast, our approach is fully autoregressive similar to decoder-only LLMs [37, 38, 5], which eliminates the need for using heterogeneous modules to process the historical and future time steps and achieves higher efficiency in terms of data and parameters via simpler architectural design.

### 2.2  Patching Operations in Transformers

The application of patches in Transformer models has demonstrated significant potential across various data modalities. For instance, BERT [12] employs subword tokenization [40] for natural language processing, while ViT [13] segments images into 2D patches for visual understanding. The patching design has also found applications in time-series forecasting [51, 57, 34], aiming at retaining local semantics and reducing computational complexity [34]. Moreover, it has shown the effectiveness in self-supervised learning, which has significantly facilitated representation learning and contributed to excellent fine-tuning results on large datasets [2, 21, 3]. Since the task of agent simulation also involves time-series data, we expect the patching mechanism to help models effectively capture the spatial-temporal interactions in driving scenarios and enhance the realism of the generated motion. Our proposed Next-Patch Prediction Paradigm (NP3) utilizes patch-level tokens in autoregressive modeling and trains each token to generate the next patch that comprises multi-step motions, which shares some similarities to multi-token prediction in LLMs [17].

## 3  Methodology

This section presents the proposed BehaviorGPT for multi-agent behavior simulation, with Figure 2 illustrating the overall framework. To begin with, we provide the formulation of our map-conditioned, multi-agent autoregressive modeling. Then, we detail the architecture of BehaviorGPT, which adopts a Transformer decoder with a triple-attention mechanism to operate sequences at the patch level. Finally, we present the objective for model training.

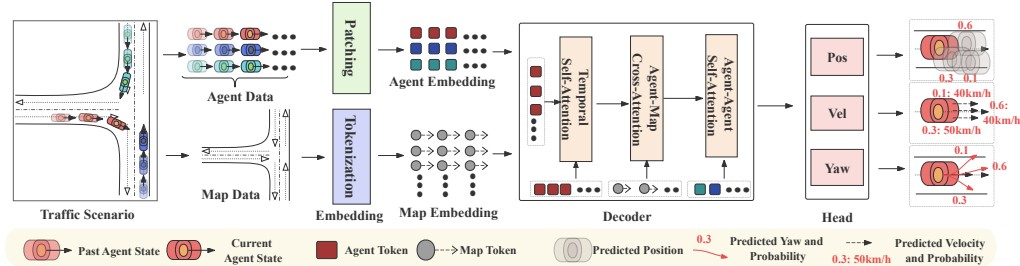

Figure 2: **Overview of BehaviorGPT.** The model takes as input the agent trajectories and the map elements, which are converted into the embeddings of trajectory patches and map polyline segments, respectively. These embeddings are fed into a Transformer decoder for autoregressive modeling based on next-patch prediction, in which the model is trained to generate the positions, velocities, and yaw angles of trajectory patches.

## 3.1 Problem Formulation

In multi-agent traffic simulation, we aim to simulate agents' future behavior in dynamic and complex environments. Specifically, we define a scenario as the composite of a vector map $M$ and the states of $N_{\text{agent}}$ agents over $T$ time steps. At each time step, the state of the $i$-th agent $S_i$ includes the agent's position, velocity, yaw angle, and bounding box size. The semantic type of agents (e.g., vehicles, pedestrians, and cyclists) are also available. Given the sequential nature of agent trajectories, we formulate the problem as sequential predictions over trajectory patches, where the prediction of each patch will affect the subsequent patches. We define an agent-level trajectory patch as

$$P_i^\tau = S_i^{((\tau-1)\times\ell+1):(\tau\times\ell)} , \ i \in \{1,\dots,N_{\text{agent}}\} , \ \tau \in \{1,\dots,N_{\text{patch}}\} , \tag{1}$$

where $\ell$ is the number of time steps covered by a patch, $N_{\text{patch}} = T/\ell$ indicates the number of patches, and $P_i^\tau$ represents the $\tau$-th trajectory patch of the $i$-th agent, with $S_i^{((\tau-1)\times\ell+1):(\tau\times\ell)}$ denoting the states within the patch. On top of $P_i^\tau$, we use $P^\tau = S_{1:N_{\text{agent}}}^{((\tau-1)\times\ell+1):(\tau\times\ell)}$ to denote the $\tau$-th multi-agent patch, where $P^\tau$ incorporates all agents' states at the $\tau$-th patch. Next, we factorize the multi-agent joint distribution over patches along the time axis according to the chain rule:

$$\Pr\left(S_{1:N_{\text{agent}}}^{1:T} \mid M\right) = \prod_{\tau=1}^{N_{\text{patch}}} \Pr\left(P^\tau \mid P^{1:(\tau-1)}, M\right) , \tag{2}$$

where $\Pr(S_{1:N_{\text{agent}}}^{1:T} \mid M)$ is the joint distribution of all agents' states over all time steps conditioned on the map $M$. Further, we factorize over agents the conditional distribution of multi-agent patches based on the assumption that agents plan their motions independently within the horizon of a patch:

$$\Pr\left(P^\tau \mid P^{1:(\tau-1)}, M\right) = \prod_{i=1}^{N_{\text{agent}}} \Pr\left(P_i^\tau \mid P^{1:(\tau-1)}, M\right) . \tag{3}$$

Considering the multimodality of agents' behavior within the horizon of a patch, we assume $\Pr(P_i^\tau \mid P^{1:(\tau-1)}, M)$ to be a mixture model consisting of $N_{\text{mode}}$ modes:

$$\Pr\left(P_i^\tau \mid P^{1:(\tau-1)}, M\right) = \sum_{k=1}^{N_{\text{mode}}} \pi_{i,k}^\tau \Pr\left(P_{i,k}^\tau \mid P^{1:(\tau-1)}, M\right) , \tag{4}$$

where $\pi_{i,k}^\tau$ is the probability of the $k$-th mode. Given the sequential nature of the states within a patch, we further conduct factorization over the states per mode using the chain rule:

$$\Pr\left(P_{i,k}^\tau \mid P^{1:(\tau-1)}, M\right) = \prod_{t=(\tau-1)\times\ell+1}^{\tau\times\ell} \Pr\left(S_{i,k}^t \mid S_{i,k}^{((\tau-1)\times\ell+1):(t-1)}, P^{1:(\tau-1)}, M\right) . \tag{5}$$

Such an autoregressive formulation can be interpreted as planning the patch-level behavior of each agent independently (Eq. (3)), freezing agents' behavior mode per $\ell$ time steps (Eq. (4)),

and autoregressively unrolling the next state under a specific behavior mode (Eq. (5)). Under this formulation, we can flexibly adjust the replan frequency during inference to control the reactivity of agents. For example, we can let agents execute $\alpha \in \{1, \ldots, \ell\}$ steps of the planned motions and choose a new behavior mode after $\alpha$ steps to react to the change in environments.

## 3.2 Relative Spacetime Representation

In our autoregressive formulation, we treat each trajectory patch as the "current" patch that is responsible for estimating the next-patch distribution during training, contrasting many existing approaches that designate one current time step per sequence [52, 49, 23]. As a result, it is inefficient to employ the well-established agent- or polyline-centric representation from the field of motion forecasting [46, 59, 33, 42, 25, 54, 43], given that these representations are computed under the reference frames determined by one current time step per sequence. For this reason, we adopt the relative spacetime representation introduced in QCNet [58] to model the patches symmetrically in space and time, achieving simultaneous multi-agent prediction when implementing Eq. (3) and allowing parallel next-patch prediction for the modeling of Eq. (2). Under this representation, the features of each map element and agent state are derived from coordinate-independent attributes, e.g., the semantic category of a map element and the speed of an agent state. On top of this, we effectively maintain the spatial-temporal relationships between input elements via relative positional embeddings. Specifically, we use $i$ and $j$ to index two different input elements and compute the relative spatial-temporal embedding by

$$\mathcal{R}_{j \to i} = \text{MLP}\left(\|\boldsymbol{d}_{j \to i}\|, \angle\left(\boldsymbol{n}_i, \boldsymbol{d}_{j \to i}\right), \Delta\boldsymbol{\theta}_{j \to i}, \Delta\boldsymbol{z}_{j \to i}, \Delta\boldsymbol{\tau}_{j \to i}\right), \tag{6}$$

where $R_{j \to i}$ is the relational embedding from $j$ to $i$, $\|d_{j \to i}\|$ is the Euclidean distance between them, $\angle(n_i, d_{j \to i})$ is the angle between $n_i$ (i.e., the orientation of $i$) and $d_{j \to i}$ (i.e., the displacement vector from $j$ to $i$), $\Delta\theta_{j \to i}/\Delta z_{j \to i}$ is the relative yaw/height from $j$ to $i$, and $\Delta\tau_{j \to i}$ is the time difference.

## 3.3 Map Tokenization and Agent Patching

Before performing spatial-temporal relational reasoning among the input elements of a traffic scenario, we must convert the raw information into high-dimensional embeddings. We first embed map information by sampling points along map polylines every 5 meters and tokenizing the semantic category of each 5-meter segment (e.g., lane centerlines, road edges, and crosswalks) via learnable embeddings. The $i$-th polyline segment's embedding is denoted by $\hat{M}_i$, which does not include any information about coordinates. On the other hand, we process agent states using attention-based patching to obtain patch-level embeddings of trajectories. For the $i$-th agent's state $S_i^t$ at time step $t$, we employ an MLP to transform the speed, the velocity vector's angle relative to the bounding box's heading, the size of the bounding box, and the semantic type of the agent, into a feature vector $\hat{S}_i^t$. To further acquire patch embeddings, we collect the feature vectors of $\ell$ consecutive agent states and apply the attention mechanism with relative positional embeddings to them:

$$\hat{P}_i^\tau = \text{MHSA}(Q = \hat{S}_i^{\tau \times \ell}, K = V = \{[\hat{S}_i^t, \mathcal{R}_i^{t \to (\tau \times \ell)}]\}_{t \in \{(\tau-1) \times \ell+1, \ldots, \tau \times \ell-1\}}), \tag{7}$$

where $\hat{P}_i^\tau$ is the patch embedding of the $i$-th agent at the $\tau$-th patch, $\text{MHSA}(\cdot)$ denotes the multi-head self-attention [47], $[:, :]$ denotes concatenation, and $\mathcal{R}_i^{t \to (\tau \times \ell)}$ indicates the positional embedding of $S_i^t$ relative to $S_i^{\tau \times \ell}$ computed according to Eq. (6). Such an operation can be viewed as aggregating the features of $S_i^{((\tau-1) \times \ell+1):(\tau \times \ell-1)}$ into $\hat{S}_i^{\tau \times \ell}$ and using the embeddings fused with high-level semantics as the agent tokens in the subsequent modules.

## 3.4 Triple-Attention Transformer Decoder

After obtaining map tokens and the patch embeddings of agents, we employ a Transformer decoder [47] with the triple-attention mechanism to model the spatial-temporal interactions among scene elements. As illustrated in Figure 3, the triple-attention mechanism considers three distinct sources of relations in the scene, including the temporal dependencies over the trajectory patches per agent, the regulations of the map elements on the agents, and the social interactions among agents.

**Temporal Self-Attention.** This module captures the relationships among the trajectory patches of each individual agent. Similar to decoder-only LLMs [37, 38, 5], it leverages the multi-head

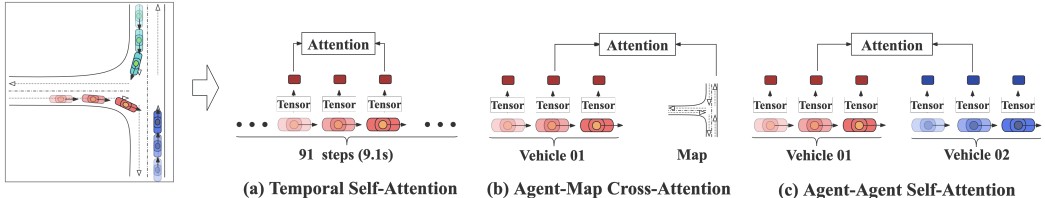

Figure 3: **Triple Attention** applies attention mechanisms to model (a) agents' sequential behaviors, (b) agents' relationships with the map context, and (c) the interactions among agents.

self-attention (MHSA) with a causal mask to enforce each trajectory patch to only attend to the preceding patches of the same agent, accommodating our autoregressive formulation. The temporal MHSA is equipped with relative positional embeddings:

$$F_{a2t,i}^{\tau} = \text{MHSA}(Q = \hat{P}_i^{\tau}, K = V = \{[\hat{P}_i^t, \mathcal{R}_i^{(t \times \ell) \to (\tau \times \ell)}]\}_{t \in \{1, \dots, \tau-1\}}), \quad (8)$$

where $F_{a2t,i}^{\tau}$ and $\hat{P}_i^{\tau}$ are the temporal-aware feature vector and the patch embedding of the $i$-th agent at the $\tau$-th patch, respectively, and $\mathcal{R}_i^{t \times \ell \to \tau \times \ell}$ embeds the relative position from $S_i^{t \times \ell}$ to $S_i^{\tau \times \ell}$, which represents the spatial-temporal relationship between the patches $P_i^t$ and $P_i^{\tau}$.

**Agent-Map Cross-Attention.** Unlike natural language which only has a sequence dimension, we must also conduct spatial reasoning to consider the environmental influence on agents' behavior. To facilitate the modeling of agent-map interactions, we apply the multi-head cross-attention (MHCA) to each trajectory patch in the scenario. Considering that a scenario may comprise an explosive number of map polyline segments and that an agent would not be influenced by map elements far away, we filter the key/value map elements in MHCA using the k-nearest neighbors algorithm [42, 54]. The agent-map cross-attention is formulated as

$$F_{a2m,i}^{\tau} = \text{MHCA}(Q = F_{a2t,i}^{\tau}, K = V = \{[\hat{M}_j, \mathcal{R}_{j \to i}^{\tau \times \ell}]\}_{j \in \mathcal{N}(i,\tau)}), \quad (9)$$

where $F_{a2m,i}^{\tau}$ is the map-aware feature vector for the $i$-th agent at the $\tau$-th patch, $\hat{M}_j$ is the embedding of the $j$-th map polyline segment, $\mathcal{R}_{j \to i}^{\tau \times \ell}$ is the relative positional embedding between the agent state $S_i^{\tau \times \ell}$ and the $j$-th map polyline segment, and $\mathcal{N}(i, \tau)$ denotes the k-nearest map neighbors of $S_i^{\tau \times \ell}$.

**Agent-Agent Self-Attention.** We further capture the social interactions among agents by applying the MHSA to the space dimension of the trajectory patches. In this module, we also utilize the locality assumption induced by the k-nearest neighbor selection for better computational and memory efficiency. Specifically, the map-aware features of trajectory patches are refined by

$$F_{a2a,i}^{\tau} = \text{MHSA}(Q = F_{a2m,i}^{\tau}, K = V = \{[F_{a2m,j}^{\tau}, \mathcal{R}_{j \to i}^{\tau \times \ell}]\}_{j \in \mathcal{N}(i,\tau)}), \quad (10)$$

where $F_{a2a,i}^{\tau}$ is the feature vector enriched with spatial interaction information among agents for the $i$-th agent at the $\tau$-th patch, $\mathcal{R}_{j \to i}^{\tau \times \ell}$ contains the relative information between the $i$-th and the $j$-th agent at the $\tau$-th patch, and $\mathcal{N}(i, \tau)$ filters the k-nearest agent neighbors of $S_i^{\tau \times \ell}$.

**Overall Decoder Architecture.** Each of the attention layers above is enhanced by commonly used components in Transformers [47], including feed-forward networks, residual connections [19], and Layer Normalization [1] in a pre-norm fashion. To enable higher-order relational reasoning, we stack multiple triple-attention blocks by interleaving the three Transformer layers. We denote the ultimate feature of the $i$-th agent at the $\tau$-th patch as $F_i^{\tau}$, which will serve as the input of the prediction head for next-patch prediction modeling.

### 3.5 Next-Patch Prediction Head

Given the interaction-aware patch features output by the Transformer decoder, we develop a next-patch prediction head to model the marginal multimodal distribution of agent trajectories, which estimates the distributional parameters of each patch's successor.

The following describes the process of next-patch prediction regarding the $\tau$-th patch of the $i$-th agent. Based on the attention output $F_i^\tau$, we intend to estimate the parameters of the next patch's mixture model pre-defined with $N_{\text{mode}}$ modes. First, we use an MLP to transform $F_i^\tau$ into $\pi_i^{\tau+1} \in \mathbb{R}^{N_{\text{mode}}}$, the mixing coefficient of the modes. In each mode, the conditional distribution of the next agent state, as depicted in Eq. (5), is considered a multivariate marginal distribution that parameterizes the position and velocity components as Laplace distributions and the yaw angle as a von Mises distribution. Based on this formulation, we employ a GRU-based autoregressive RNN [9] to unroll the states within the next patch step by step, with each step being conditioned on the previously predicted states. Specifically, The hidden state $h_{i,k}^{\tau,t}$ of the RNN is initialized with $F_i^\tau$ at $t = 1$ for $\forall k \in \{1, \ldots, N_{\text{modes}}\}$. At each step of the rollout, we use an MLP to estimate the location and scale parameters of the next agent state's position and velocity based on the hidden state. On the other hand, the MLP also estimates the location and concentration parameters of the next yaw angle. The location parameters of the newly predicted state, including the 3D positions, the 2D velocities, and the yaw angle, are used to update the RNN's hidden state directly without relying on the predicted scale/concentration parameters for sampling. The whole process is summarized as follows:

$$
\begin{aligned}
\pi_{i,k}^{\tau+1} &= \text{MLP}([F_i^\tau, Z_k]) , \\
h_{i,k}^{\tau,1} &= F_i^\tau , \\
\mu_{i,k}^{\tau \times \ell + t}, \ b_{i,k}^{\tau \times \ell + t}, \ \kappa_{i,k}^{\tau \times \ell + t} &= \text{MLP}([h_{i,k}^{\tau,t}, Z_k]) , \\
h_{i,k}^{\tau,t+1} &= \text{RNN}(h_{i,k}^{\tau,t}, \ \text{MLP}(\mu_{i,k}^{\tau \times \ell + t})) ,
\end{aligned}
\tag{11}
$$

where $\{\mu_{i,k}^{\tau \times \ell + t} \in \mathbb{R}^6\}_{t \in \{1, \ldots, \ell\}}$, $\{b_{i,k}^{\tau \times \ell + t} \in \mathbb{R}^5\}_{t \in \{1, \ldots, \ell\}}$, and $\{\kappa_{i,k}^{\tau \times \ell + t} \in \mathbb{R}\}_{t \in \{1, \ldots, \ell\}}$ are the location, scale, and concentration parameters in the $k$-th mode, and $Z_k$ is the $k$-th learnable mode embedding.

### 3.6 Training Objective

To train BehaviorGPT, we apply the negative log-likelihood loss $\mathcal{L}_{\text{NLL}}$ to the factorized distribution of $\Pr(S_{1:N_{\text{agent}}}^{1:T} \mid M)$ as formulated previously:

$$
\mathcal{L}_{\text{NLL}} = \sum_{\tau=1}^{N_{\text{patch}}} \sum_{i=1}^{N_{\text{agent}}} - \log \sum_{k=1}^{N_{\text{mode}}} \pi_{i,k}^\tau \prod_{t=(\tau-1) \times \ell + 1}^{\tau \times \ell} \Pr\left(S_{i,k}^t \mid S_{i,k}^{((\tau-1) \times \ell + 1):(t-1)}, P^{1:(\tau-1)}, M\right) .
\tag{12}
$$

Note that each ground-truth trajectory patch is transformed into the viewpoint of its previous patch. During training, we utilize teacher forcing to parallelize the modeling of next-patch prediction and ease the learning difficulty, but we do not use the ground-truth agent states when updating the RNN's hidden states, intending to train the model to recover from its mistakes made in next-state prediction.

## 4 Experiments

This section first introduces the dataset and the evaluation metrics used in our experiments, followed by presenting the implementation details and the rollout results obtained by BehaviorGPT on the Waymo Open Sim Agents Benchmark [32]. Finally, we conduct ablation studies to further compare and analyze the performance of BehaviorGPT under various settings.

### 4.1 Dataset and Metrics

Our experiments are conducted on the Waymo Open Motion Dataset (WOMD) [15]. The dataset comprises 486,995/44,097/44,920 training/validation/testing scenarios. Each scenario includes 91-step observations sampled at 10 Hz, totaling 9.1 seconds. Given 11-step initial states of the scenarios, we simulate up to 128 agents and generate 80 simulation steps per agent at 0.1-second intervals in an autoregressive and reactive manner. Each agent requires 32 simulations comprising x/y/z centroid coordinates and a heading value. The results on the test set are obtained by utilizing the full training set, while the performance on the validation set is based on 20% of training data unless specified.

We use various metrics for evaluation. The minADE measures the minimum average displacement error over multiple simulated trajectories, assessing trajectory accuracy. REALISM is the meta-metric

Table 1: Test set results in the 2024 Waymo Open Sim Agents Challenge.

| Model | #Param | minADE (↓) | REALISM (↑) | LINEAR SPEED (↑) | LINEAR ACCEL (↑) | ANG SPEED (↑) | ANG ACCEL (↑) | DIST TO OBJ (↑) | COLLISION (↑) | TTC (↑) | DIST TO ROAD EDGE (↑) | OFFROAD (↑) |
|---|---|---|---|---|---|---|---|---|---|---|---|---|
| Linear Extrapolation [32] | - | 7.5148 | 0.3985 | 0.0434 | 0.1661 | 0.2522 | 0.4393 | 0.2154 | 0.3905 | 0.7555 | 0.4801 | 0.4426 |
| TrafficBotsV1.5 [55] | 10M | 1.8825 | 0.6988 | 0.3361 | 0.3497 | 0.4512 | 0.5844 | 0.3596 | 0.8083 | 0.8209 | 0.6423 | 0.9134 |
| VBD [23] | 12M | 1.4743 | 0.7200 | 0.3591 | 0.3664 | 0.4197 | 0.5222 | 0.3683 | 0.9341 | 0.8153 | 0.6508 | 0.8788 |
| MVTE [49] | >65M | 1.6770 | 0.7302 | 0.3506 | 0.3531 | 0.4974 | 0.6000 | 0.3743 | 0.9049 | **0.8310** | 0.6655 | 0.9071 |
| GUMP [22] | 523M | 1.6041 | 0.7431 | 0.3567 | **0.4111** | **0.5089** | **0.6353** | 0.3707 | 0.9403 | 0.8276 | 0.6686 | 0.9028 |
| **BehaviorGPT (Ours)** | 3M | **1.4147** | **0.7473** | **0.3615** | 0.3365 | 0.4806 | 0.5544 | **0.3834** | **0.9537** | 0.8308 | **0.6702** | **0.9349** |

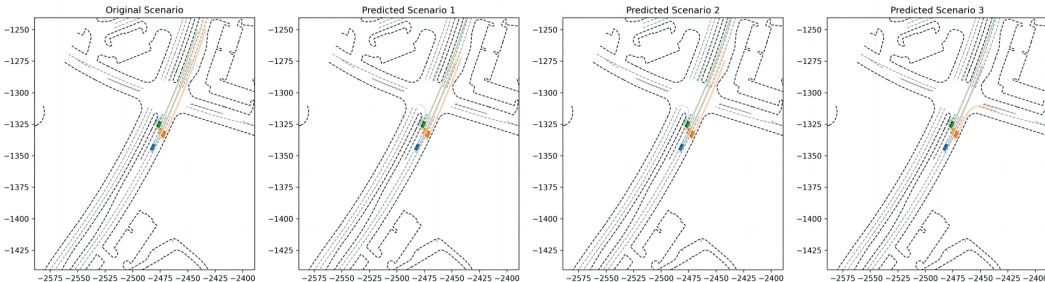

Figure 4: High-quality simulations produced by BehaviorGPT, where multimodal behaviors of agents are simulated realistically.

that expects the simulations to match the real-world distribution. LINEAR SPEED and LINEAR ACCEL evaluate the realism regarding speed and acceleration. Similarly, ANG SPEED and ANG ACCEL measure the realism of angular speed and acceleration. DIST TO OBJ considers the distances to objects, while COLLISION and TTC assess the simulation performance in terms of collision and time to collision. Finally, DIST TO ROAD EDGE and OFFROAD focus on map compliance.

## 4.2 Implementation Details

The optimal patch size we experimented with is 10, corresponding to 1 second. All hidden sizes are set to 128. Each attention layer has 8 attention heads with 16 dimensions per head. To save training resources, we limit the maximum number of agents per scenario to 128 and restrict the maximum number of neighbors in kNN attention layers to 32. The prediction head produces 16 modes per agent and time step. We train the models for 30 epochs on 8 NVIDIA RTX 4090 GPUs with a batch size of 24, utilizing the AdamW optimizer [31]. The weight decay rate and dropout rate are both set to 0.1. The learning rate is initially set to $5 \times 10^{-4}$ and decayed to 0 following a cosine annealing schedule [30]. Our results in the 2024 WOSAC are obtained using a single model with 2 decoding blocks and a total of 3M parameters. To produce 32 replicas of rollouts, we randomly sample behavior modes from agents' next-patch distributions until completing the 8-second multi-agent trajectories, and we repeat this process with different random seeds. The final results on the leaderboard are based on a replan rate of 2 Hz, while the ablation studies are based on a 1-Hz replan rate unless specified.

## 4.3 Quantitative Results

We report the test set results in Table 1. Notably, BehaviorGPT achieves the lowest minADE and the best REALISM, underscoring the model's ability to match the real-world distribution. Its excellent performance on COLLISION and OFFROAD also indicates that the model has successfully captured the agent-agent and agent-map interactions in driving scenarios. Besides the benchmarking results, we also compare the number of model parameters in BehaviorGPT and other baselines. Table 1 demonstrates that BehaviorGPT, with only 3M parameters, achieves more realistic simulation than significantly larger models like MVTE [49] and GUMP [22], which demonstrates the parameter efficiency of our approach. Without employing tricks like data augmentation, model ensemble, or post-processing steps, BehaviorGPT won first place in the 2024 Waymo Open Sim Agents Challenge.

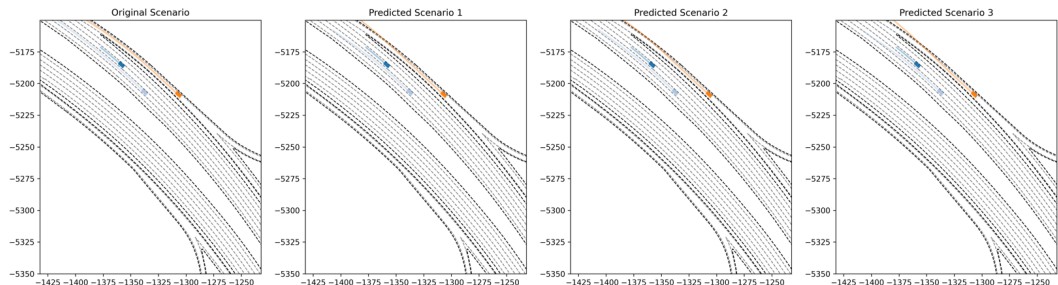

Figure 5: A typical failed case produced by BehaviorGPT, where offroad trajectories are generated owing to the compounding error caused by autoregressive modeling.

## 4.4 Qualitative Results

Figure 4 visualizes some qualitative results of the rollouts produced by our model. In this scenario, BehaviorGPT can generate multiple plausible futures given the same initial states of agents, which demonstrates its capability of simulating diverse yet realistic agent behavior. However, we also note that autoregressive models still suffer from accumulated errors in some cases. As shown in Figure 5, the vehicle in orange gradually goes out of the road as time goes by, which indicates the inherent limitations of autoregressive generation.

## 4.5 Ablation Studies

We conduct some ablation studies to gain a more in-depth understanding of our approach.

**Impact of patch size.** Table 2 presents the results of BehaviorGPT with varying patch sizes. According to the results, it is evident that using a patch size of 5, i.e., training and predicting with 2-Hz tokens, significantly outperforms the baseline without patching. Moreover, increasing the patch size to 10 further enhances the overall performance. These results demonstrate the benefits of incorporating the NP3 into agent simulation. However, changing the patch size also leads to a variation in replan frequency, which also has an influence on simulation. Next, we investigate the impact of replan frequency on the test set using the model submitted to the 2024 WOSAC.

Table 2: Impact of patch size on the validation set.

| Patch Size | Replan Frequency | minADE (↓) | REALISM (↑) | LINEAR SPEED (↑) | LINEAR ACCEL (↑) | ANG SPEED (↑) | ANG ACCEL (↑) | DIST TO OBJ (↑) | COLLISION (↑) | TTC (↑) | DIST TO ROAD EDGE (↑) | OFFROAD (↑) |
|---|---|---|---|---|---|---|---|---|---|---|---|---|
| 1 | 10 Hz | 2.3752 | 0.6783 | 0.2559 | 0.2088 | 0.4022 | 0.5094 | 0.3201 | 0.9002 | 0.8015 | 0.6149 | 0.8432 |
| 5 | 2 Hz | 1.5599 | 0.7273 | **0.3543** | **0.3218** | 0.4623 | **0.5435** | 0.3768 | 0.9181 | **0.8339** | 0.6564 | 0.9077 |
| 10 | 1 Hz | **1.5203** | **0.7335** | 0.3517 | 0.3023 | **0.4734** | 0.5432 | **0.3797** | **0.9358** | 0.8329 | **0.6645** | **0.9132** |

**Impact of replan frequency.** During inference, we vary the replan frequency of the model with a patch size of 10 by discarding a portion of the predicted states at each simulation step. As shown in Table 3, increasing the replan frequency from 1 Hz to 2 Hz can even improve the overall performance, which may benefit from the enhanced reactivity. This phenomenon demonstrates that the performance gain is not merely due to the lower replan frequency, as the model with a patch size of 10 beats that with a patch size of 5 even harder if using the same replan frequency of 2 Hz. However, using an overly high replan frequency harms the performance, as indicated by the third row of Table 3. Overall, we conclude that using a larger patch indeed helps long-term reasoning, but a moderate replan frequency is important for temporal stability, which may be neglected by prior works.

Table 3: Impact of replan frequency on the test set.

| Patch Size | Replan Frequency | minADE (↓) | REALISM (↑) | LINEAR SPEED (↑) | LINEAR ACCEL (↑) | ANG SPEED (↑) | ANG ACCEL (↑) | DIST TO OBJ (↑) | COLLISION (↑) | TTC (↑) | DIST TO ROAD EDGE (↑) | OFFROAD (↑) |
|---|---|---|---|---|---|---|---|---|---|---|---|---|
| 10 | 1 Hz | 1.5405 | 0.7414 | 0.3553 | 0.3153 | 0.4695 | 0.5303 | 0.3772 | 0.9520 | 0.8285 | 0.6664 | 0.9308 |
| 10 | 2 Hz | **1.4147** | **0.7473** | **0.3615** | 0.3365 | **0.4806** | 0.5544 | **0.3834** | **0.9537** | **0.8308** | **0.6702** | **0.9349** |
| 10 | 5 Hz | 1.5693 | 0.7342 | 0.3430 | **0.3472** | 0.4663 | **0.5673** | 0.3722 | 0.9429 | 0.8253 | 0.6534 | 0.9089 |

**Impact of multi-agent interaction modeling.** We remove all agent-agent self-attention layers in the first row of Table 4 to show that modeling the interactions among agents can boost minADE and REALISM. In particular, the realism in terms of collision is improved by $34.66\%$ when employing agent-agent self-attention.

Table 4: Impact of agent-agent self-attention on the validation set.

| Agent-Agent Self-Attention | minADE (↓) | REALISM (↑) | DIST TO OBJ (↑) | COLLISION (↑) | TTC (↑) |
|---|---|---|---|---|---|
| ✗ | 2.1489 | 0.6659 | 0.3539 | 0.6987 | 0.8070 |
| ✓ | **1.6247** | **0.7349** | **0.3783** | **0.9409** | **0.8320** |

Table 5: Effects of training data on the validation set.

| Train Data | #Param | minADE (↓) | REALISM (↑) |
|---|---|---|---|
| 20% | 5M | 1.4881 | 0.7396 |
| 50% | 5M | 1.4060 | 0.7427 |
| 100% | 5M | **1.3804** | **0.7438** |

Table 6: Effects of model depth on the validation set.

| Model Depth | #Param | minADE (↓) | REALISM (↑) |
|---|---|---|---|
| 2 | 3M | 1.6247 | 0.7349 |
| 3 | 4M | 1.5381 | 0.7387 |
| 4 | 5M | **1.4881** | **0.7396** |

Table 7: Effects of model width on the validation set.

| Model Width | #Param | minADE (↓) | REALISM (↑) |
|---|---|---|---|
| 64 | 800K | 1.9637 | 0.7251 |
| 128 | 3M | 1.6247 | 0.7349 |
| 192 | 7M | **1.4993** | **0.7382** |

Table 8: Extrapolation ability to generate longer sequences.

| Training | Inference | minADE (↓) | REALISM (↑) | LINEAR SPEED (↑) | LINEAR ACCEL (↑) | ANG SPEED (↑) | ANG ACCEL (↑) | DIST TO OBJ (↑) | COLLISION (↑) | TTC (↑) | DIST TO ROAD EDGE (↑) | OFFROAD (↑) |
|---|---|---|---|---|---|---|---|---|---|---|---|---|
| 9.1 sec | 9.1 sec | **1.6247** | **0.7349** | 0.3546 | 0.3105 | **0.4689** | **0.5363** | 0.3783 | **0.9409** | **0.8320** | **0.6605** | **0.9163** |
| 5.0 sec | 9.1 sec | 1.6294 | 0.7333 | **0.3565** | **0.3471** | 0.4613 | 0.5293 | **0.3813** | 0.9375 | 0.8273 | 0.6585 | 0.9100 |

**Scaling with data.** We train our models with different proportions of training data. All the models have 4 decoding blocks and a hidden size of 128, totaling 5M parameters. As shown in Table 5, BehaviorGPT is able to achieve remarkable performance with merely $20\%$ of training data, which is attributed to the high data efficiency of our approach. Increasing the proportion of training data from $20\%$ to $50\%$ further improves the performance on minADE and REALISM, and training on $100\%$ of the data continues to gain enhancement. Judging from the trend in Table 5, we believe that feeding more data for model training will continuously achieve better simulation performance.

**Scaling with model size.** We investigate the effects of scaling up the model size based on some preliminary experiments with $20\%$ of training data. In Table 6, we vary the number of decoding blocks while fixing the hidden size as 128. On the other hand, we fix the number of decoding blocks as 2 and vary the hidden size, as depicted in Table 7. Based on the experimental results, we can summarize that enlarging the model consistently leads to more realistic simulation, which showcases the potential of BehaviorGPT for scaling up.

**Extrapolation ability.** We tried training a model on 5-second sequences and generating 9.1-second sequences during inference. The results in Table 8 show that this model achieves similar performance compared with the baseline trained with 9.1-second sequences, demonstrating our approach's extrapolation ability to generate longer sequences.

## 5 Conclusion

This work introduced BehaviorGPT, a fully autoregressive architecture designed to enhance smart agent simulation for autonomous driving. By applying homogeneous Transformer blocks to entire trajectory snippets and utilizing relative spacetime representations, BehaviorGPT simplifies the modeling process and maximizes data utilization. To enable high-level understanding and long-range interaction reasoning in space and time, we developed the Next-Patch Prediction Paradigm, which tasks models with generating trajectory patches instead of single-step states. Experimental results on the Waymo Open Sim Agents Challenge demonstrate that BehaviorGPT achieves outstanding performance with merely 3M model parameters, highlighting its potential to further improve the realism of agent simulation with more data and computation.

**Limitations.** First, BehaviorGPT is currently inferior in kinematics-related performance, which can be enhanced by incorporating a kinematic model, e.g., the bicycle model. Second, the current version of BehaviorGPT does not support controlling agent behavior with specific prompts such as language and goal points. However, achieving controllable generation should be trivial given a powerful base model. Finally, we have not verified whether BehaviorGPT will facilitate the development of motion planning, which we leave as future work.

## Acknowledgement

This project is supported by a grant from Hong Kong Research Grant Council under GRF project 11216323 and CRF C1042-23G.

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
