# OpenReview forum: "BehaviorGPT: Smart Agent Simulation for Autonomous Driving with Next-Patch Prediction"
_NeurIPS.cc/2024/Conference — NeurIPS 2024 poster_

### Official Review · Reviewer_gNhc · 2024-06-27

**Soundness:** 3
**Presentation:** 2
**Contribution:** 3
**Rating:** 5
**Confidence:** 4

**Summary:**

BehaviorGPT: Smart Agent Simulation for Autonomous Driving with Next-Patch Prediction introduces a model architecture for multi-agent simulation of dynamic traffic actors.  Improved simulation capabilities are essential to the safe and rapid development of autonomous vehicles.  BehaviorGPT structures multi-agent simulation a next-patch-predition problem.  Given a history of trajectory patches (embeddings of trajectory subsequences of fixed length), the model predicts the next patch's set of states for each actor independently.  These predictions can then be treated as a patch and the process can be repeated in an autoregressive rollout. Through use of relative encodings (in both time and space) and a "decoder-only" architecture, BehaviorGPT can be trained in parallel any continuous subset of patches.  BehaviorGPT is state-of-the-art on the Waymo Sim Agents Benchmark with impressive scores for both trajectory accuracy (minADE) as well as realism.  Furthermore, BehaviorGPT accomplishes these SOTA results with only 3M parameters (an order of magnitude lower than other competing approaches).

**Strengths:**

- The paper has several empirical strengths.  Most notably it is SOTA on a challenging benchmark (the Waymo Sim Agents Benchmark) despite being an surprisingly small model (3M parameters).  Improvement on this benchmark is meaningful as simulation (and the existing sim2real gap) is a major challenge for development and deployment of autonomous vehicles.

- The manuscript does an excellent job of contextualizing this work within the growing cannon of motion forecasting and simulation literature.

- The manuscript does a nice job providing intuitive motivations for their architectural choices.  For example, the RNN in the prediction head is clearly motivated by the sequential nature of trajectory simulation.

**Weaknesses:**

At a very high level, I think this paper presents a model which produces impressive results on an important benchmark (which represents an important problem).  However, I think that the current manuscript falls short in multiple dimensions.  I do not expect or think that the authors must make substantive changes with respect to all of the points below.  However if the authors were able to expand or address a couple of the weaknesses described below it would significantly improve the strength of the submission.

## 1. Scope down claims slightly
The results are super impressive, but I would scope down some of the adjacent claims.  For example, there is the assertion that there is a big sample efficiency gain over MTVA and Trajeglish but there is no evidence to support that.  I also think that the claim of being the first "decoder only" model is a bit overstated; this is largely semantic.  Other authors writing the same paper may have called your decoder the encoder and the prediction head the decoder.  Additionally the paper Query-Centric Trajectory Prediction had an awesome ablation in their paper where the reduced the encoder to zero layers and still had excellent results).  Instead of focusing on being decoder only, focus on the ability for further parallelize training.  I also find the statements around "next patch prediction" to be a bit strange because you predict states not a patch ... it only becomes a patch when it is embedded by the decoder.

## 2. Reproducibility
I don't find the current manuscript to be sufficiently detailed to reproduce the model.  (Code release will fix a lot of this).  Some specific examples include:
- More details are need around map tokenization - I would not know how to translate your text into code here.
- In the prediction head it is not clear to me if there is a single RNN and a single MLP which is outputing all the mixture components or there are multiple copies (I assume the former ... but its not clear).
- Embedding sizes?

## 3. Manuscript quality
- The figures and captions could use work.  In general the captions are far to brief and do not provide enough information to a reader skimming the paper.  The figures themselves, with the exception of figure 2 provide little explanatory value.  Figure 3 is a poor use of space as most of the figure is an identical cartoon repeated three times.  It would be great to replace some of the non-useful figures with visual representations of the model output (i.e. real examples).
- References need to be cleaned up ... I only skimmed but a couple examples include [18] where Densetnt -> DenseTNT and [48] should cite the NeurIPS 2023 manuscript rather than arxiv.
- I would steer away from introducing the concept of a scene level patch $P^{\tau}$ as it is never actually constructed.

## 4. Forecasting metrics/leaderboards
As noted in the paper, the simulation problem and the forecasting problem are quite similar.  The minADE numbers suggest that this method would also perform well on motion forecasting benchmarks.  I would love to see numbers there.  I do believe the Waymo Sim Benchmark is an important one, but it is also new and has not had as many competitive submissions as the Waymo or Argoverse forecasting benchmarks.  How does this model compare with QCNet on those benchmarks?

## 5. Model introspection
It is not clear from this analysis, which elements of the architecture are actually important for its performance.  I would like to see more ablations.  What happens when we drop out one of the factorized attention modules?  Does the model really get any worse if we don't to  the a2a attention?  Does the model improve/degrade if we only do each attention once, or three times?  Is the RNN actually important?  What if we just predict with an MLP and interpolate?  Are the patches actually important or just going from 10Hz to 1Hz?

## 6. Output introspection
It is typical for papers in this space to provide analysis beyond aggregate metrics.  There should be a visual analysis of the output.  Examples should be provided of instances where the model produced excellent output (specifically in comparison to other methods).  Examples should also be provided of instances where the model produced poor outputs.  Are there any connecting themes?  I.e. does the model struggle with certain road geometries or interactions?

## 7. Alternative validation of value
The real claim of this paper is that they have produced a model which will produce more realistic simulations which in turn lead to improved development of autonomous vehicles.  Benchmarks are proxy evidence for this claim.  Can this be supported in some other way as well?  If we were to train a planner or motion forecasting method on output from this sim vs. another sim, would it perform better?

## 8. Inference performance
Autonomous vehicle companies need to run simulations at massive scales. (Tens of millions of scenarios).  To that end it would be nice to see the paper report the inference performance of their model (on reported hardware).

## Conclusion
As noted in the beginning, not all of these areas need to be addressed in the limited rebuttal window. 1-3 should definitely be addressed along with some subset of 4-8 (7 is a long shot for sure).

**Questions:**

It wasn't clear how many patches were available as history during training or inference?

**Limitations:**

No concerns.

---

> ### Author Rebuttal · Authors · 2024-08-05
>
> We sincerely appreciate your constructive suggestions, which greatly help improve the quality of our manuscript. In the following, we attempt to address some critical concerns you raised.
>
> **1. Scope down claims slightly**
>
> (1) **Sample efficiency**: We evaluate our models trained with different proportions of training data. Our model is able to achieve very decent performance when trained on merely 20% of data, which is attributed to the high sample efficiency of our approach.
> | Training Data | minADE $\downarrow$ | Realism $\uparrow$ | Offroad $\uparrow$  |
> | :---: | :---: | :---: | :---: |
> | 20% | 1.4881 | 0.7396 | 0.9207 |
> | 50% | 1.4060 | 0.7427 | 0.9250 |
> | 100% | 1.3804 | 0.7438 | 0.9268 |
>
> (2) **Decoder-only**: We agree that the notion of "decoder-only" is somewhat semantic. We will revise the introduction of our paper to emphasize our approach's capability of parallel training instead of claiming "the first decoder-only agent simulator."
>
> (3) **Next-patch prediction**: In our approach, each token in the decoder has fused the information of multi-step states and takes charge of generating multiple subsequent states; since we define multiple consecutive states as a patch, we think it appropriate to use the term "next-patch prediction."
> ***
> **2. Reproducibility**
>
> More details on implementation will be supplemented in the revised version. Below are the ones that you are concerned about:
>
> (1) **Map tokenization**: We sample points along the polylines every 5 meters and tokenize the semantic category of each 5-meter segment via learnable embeddings. The shape of the embedding table is [17, 128], indicating 17 categories and a hidden size of 128.
>
> (2) **RNN head**: A single RNN+MLP outputs all the mixture components.
>
> (3) **Hidden size**: All hidden sizes are 128.
> ***
> **3. Manuscript quality**
>
> (1) **Figure**: Thank you for the suggestions. We will adjust the figures (e.g., removing the redundant cartoon in Figure 3 and deleting Figure 4) to leave some space for qualitative results. We will also expand the captions to describe more details.
>
> (2) **Reference**: Thank you for spotting the issues. We will clean up the references carefully.
>
> (3) **Notation**: Indeed, the notion of scene-level patches is not that important in our model. We will change Equation 2 into an inline equation.
> ***
> **4. Forecasting results**
>
> BehaviorGPT considers the **joint distribution of ALL agents in the scene**, which is misaligned with the objective of existing motion prediction benchmarks. For example, the Waymo Motion Prediction Benchmark and the Argo 2 Single-Agent Motion Forecasting Benchmark are for marginal prediction; the Waymo Interaction Prediction Benchmark and the Argo 2 Multi-Agent Motion Forecasting Benchmark concern the joint distribution of a subset of agents in the scene. Indeed, we can customize our model architecture and training objective to obtain high scores on motion forecasting benchmarks, but that is out of the scope of this work. Thus, it is more reasonable to compare our approach with joint multi-agent motion prediction models that consider the joint distribution of all agents. We also noted that QCNeXt, a joint multi-agent prediction model that has won 1st place in the CVPR 2023 Argo 2 multi-agent motion forecasting challenge, is also on the WOSAC leaderboard. We found that QCNeXt performs much better on minADE (1.08 vs our 1.54), but its closed-loop performance lags far behind most simulation-oriented models. It seems to be a trade-off between open-loop and closed-loop performance.
> ***
> **5. Model introspection**
>
> (1) **A2A attention**: Without A2A attention, the model cannot capture agent interactions.
> | A2A | minADE $\downarrow$ | Realism $\uparrow$ | Collision $\uparrow$  |
> | :---: | :---: | :---: | :---: |
> | &check; | 1.6247 | 0.7349 | 0.9409 |
> | &cross; | 2.1489 | 0.6659 | 0.6987 |
>
>
> (2) **#Layers**: We try to fix the hidden size as 128 and vary the number of attention layers. We found that increasing the depth of the models can benefit the performance.
> | #Layer | minADE $\downarrow$ | Realism $\uparrow$ | Offroad $\uparrow$  |
> | :---: | :---: | :---: | :---: |
> | 1 | 1.7318 | 0.7319 | 0.9149 |
> | 2 | 1.6247 | 0.7349 | 0.9163 |
> | 3 | 1.5381 | 0.7387 | 0.9199 |
> | 4 | 1.4881 | 0.7396 | 0.9207 |
>
> (3) **Patching**: We try to increase the replan frequency by discarding a portion of the predicted states at each simulation step. The test set results produced by the model with a patch size of 10 are shown as follows. From the table, we can see that increasing the replan frequency from 1 Hz to 2 Hz can even improve the overall performance, which may benefit from the enhanced reactivity. This phenomenon demonstrates that the performance gain is not merely due to the lower replan frequency, as the model with a patch size of 10 beats that with a patch size of 5 even harder if using the same replan frequency (i.e., 2Hz). However, we found that an overly high replan frequency harms the performance, as indicated by the third row of the table. Overall, we conclude that using a larger patch indeed helps long-term reasoning, but a moderate replan frequency is important for temporal stability.
> | Patch Size | Replan Frequency | minADE $\downarrow$ | Realism $\uparrow$ | Offroad $\uparrow$  |
> | :---: | :---: | :---: | :---: | :---: |
> | 10 | 1 Hz | 1.5405 | 0.7414 | 0.9308 |
> |10 | 2 Hz | **1.4147** | **0.7473** | **0.9349** |
> | 10 | 5 Hz | 1.5693 | 0.7342 | 0.9089 |
> ***
> **6. Output introspection**
>
> Please refer to the PDF file in the general response.
> ***
> **7. Alternative validation**
>
> Validating the usefulness of data-driven simulators for motion planning/prediction is definitely something worthy of doing. This will be our next step!
> ***
> **8. Inference**
>
> The average latency per simulation step is less than 10 ms on an NVIDIA L20 GPU (seq length: 9 secs).
> ***
> **9. Number of patches**
>
> 9-sec sequences are used for training, while 1-sec trajectories are used as initial history during inference.

---

> > ### Comment · Reviewer_gNhc · 2024-08-08
> > **Rebuttal followup**
> >
> > Thank you to the authors for their thorough rebuttal.  I look forward to reading the revised manuscript.  I have a couple of small follow up points:
> >
> > *On sample efficiency* - The claim in the paper is a comparative claim.  I.e. this method is "more efficient" than other methods.  The table provided demonstrates that this method performs well with a fraction of the training data.  Is there any reason to believe that other methods wouldn't scale similarly?
> >
> > *On inference latency* - I would love more information here.  The 10ms, how many agents is that for?  Is that a mean?  What is the variance of inference times?
> >
> > In short, the authors have addressed most of my concern and pending a revised manuscript I would significantly raise my score.

---

> > > ### Author Response · Authors · 2024-08-09
> > > **Thanks for your positive feedback!**
> > >
> > > Thank you very much for the positive feedback! We are committed to improving the quality of our manuscript based on the discussion during the rebuttal, though we are unable to upload the revised paper at this stage according to the policy of NeurIPS 2024.
> > >
> > > Regarding your follow-up questions:
> > >
> > > **1. Sample efficiency.**
> > >
> > > Currently, few or no methods on the benchmark have been open-source, which brings difficulties in replicating different models and evaluating their sample efficiency directly. Perhaps a piece of indirect evidence supporting our claim is Figure 11 in the Trajeglish paper, which indicates that the Trajeglish model is data-hungry. More direct evidence is that the 5M BehaviorGPT trained on 20% data can achieve lower minADE than the Trajeglish trained on 100%.
> > >
> > > **2. Inference efficiency.**
> > >
> > > The number is averaged over all scenarios on the validation set, with the standard variance of 3ms. The number of agents on the validation set is 65$\pm$36 (max: 128, min: 2).

---

> > > > ### Comment · Reviewer_gNhc · 2024-08-09
> > > > **Rebuttal followup**
> > > >
> > > > Thank you again to the authors for the followup.  Based on rebuttals to all reviewers and commitments for the final manuscript I would like to change my score to a 7 (accept).

---

> > > > > ### Author Response · Authors · 2024-08-12
> > > > > **Thank you**
> > > > >
> > > > > Thank you again for the valuable suggestions!

---

### Official Review · Reviewer_pxxg · 2024-07-08

**Soundness:** 3
**Presentation:** 2
**Contribution:** 3
**Rating:** 7
**Confidence:** 4

**Summary:**

This work focuses on multi-agent simulation for autonomous driving. Instead of commonly used encoder-decoder structure, the authors propose a decoder-only autoregressive architecture for better data utilization, and achieve SOTA on Waymo SimAgents Benchmark.

**Strengths:**

* The low data utilization problem using common encoder-decoder structure is identified, which requires a sequence be split into history and future. In proposed autoregressive architecture, each time step is treated as current, resulting in higher data utilization.
* Next-Patch Prediction Paradigm is introduced to force the model perform long-range interaction, preventing the shortcut learning in next-token prediction.

**Weaknesses:**

* It seems that no solid evidence is provided to prove the model performance "scales seamlessly with data and computation". I do not think Table 4 can support this assertion.
* Though parameter-efficient, it may not be superior in terms of computation or inference latency, providing more details about this would be better.

**Questions:**

See weaknesses.

**Limitations:**

See weaknesses.

---

> ### Author Rebuttal · Authors · 2024-08-02
>
> We sincerely thank your valuable feedback. To resolve your concerns, we try our best to conduct some scaling experiments under the constraints of limited time and computing budget.
>
> **Q1: Can BehaviorGPT scale with data and computation?**
>
> **A1:** We answer this question regarding the quantity of training data, the hidden size, and the number of decoder layers.
>
> (1) **Quantity of data**: We evaluate our models (5M parameters, hidden size = 128, #Decoder layer = 4) trained with different proportions of training data provided by the Waymo Open Motion Dataset. As shown in the table below, our model is able to achieve very decent performance when trained on merely 20% of training data, which is attributed to the high data efficiency of our approach. Increasing the proportion of training data from 20% to 50% further improves the performance across various metrics. Moreover, training on 100% of the data continues to make the model more powerful. Judging from the trend indicated in the table, we believe that feeding more data for model training will continuously improve the overall performance.
> | Training Data | minADE $\downarrow$ | Realism $\uparrow$ | Linear Speed  $\uparrow$  | Linear Acceleration $\uparrow$  | Offroad $\uparrow$  |
> | :---: | :---: | :---: | :---: | :---: | :---: |
> | 20% | 1.4881 | 0.7396 | 0.3633 | 0.3181 | 0.9207 |
> | 50% | 1.4060 | 0.7427 | 0.3637 | 0.3203 | 0.9250 |
> | 100% | 1.3804 | 0.7438 | 0.3655 | 0.3227 | 0.9268 |
>
> (2) **Hidden size**: We vary the hidden size to obtain models with different parameters. The experiments use 20% of the training data and 2 layers of the Transformer decoder. As depicted by the table below, increasing the hidden size consistently improves the performance.
> | Hidden Size | #Param | minADE $\downarrow$ | Realism $\uparrow$ | Linear Speed  $\uparrow$  | Collision $\uparrow$  | Offroad $\uparrow$  |
> | :---: | :---: | :---: | :---: | :---: | :---: | :---: |
> | 64 | 800K | 1.9637 | 0.7251 | 0.3369 | 0.9229 | 0.9056 |
> |128 | 3M | 1.6247 | 0.7349 | 0.3546 | 0.9409 | 0.9163 |
> |192 | 7M | 1.4993 | 0.7382 | 0.3646 | 0.9439 | 0.9185 |
>
> (3) **Number of decoder layers**: We also try to fix the hidden size as 128 and vary the number of decoder layers, obtaining models by training on 20% of the data. Based on the experimental results below, we can conclude that increasing the depth of the models can benefit the performance.
> | #Decoder Layer | #Param | minADE $\downarrow$ | Realism $\uparrow$ | Linear Speed  $\uparrow$  | Collision $\uparrow$  | Offroad $\uparrow$  |
> | :---: | :---: | :---: | :---: | :---: | :---: | :---: |
> | 1 | 2M | 1.7318 | 0.7319 | 0.3465 | 0.9319 | 0.9149 |
> | 2 | 3M | 1.6247 | 0.7349 | 0.3546 | 0.9409 | 0.9163 |
> | 3 | 4M | 1.5381 | 0.7387 | 0.3570 | 0.9450 | 0.9199 |
> | 4 | 5M | 1.4881 | 0.7396 | 0.3633 | 0.9481 | 0.9207 |
>
> It is a pity that we do not have more computing resources to experiment with even larger models, and we welcome researchers with sufficient computing resources to examine the scalability of our architecture after we contribute our code to the open-source community.
>
> ***
>
> **Q2: What about the inference latency?**
>
> **A2**: The average latency per simulation step is less than 10 ms on an NVIDIA L20 GPU (seq length: 9 secs).

---

> > ### Comment · Reviewer_pxxg · 2024-08-09
> >
> > Thank you to the authors for their detailed response, i think most of my concerns are resolved. I would like to have my rating unchanged.

---

> > > ### Author Response · Authors · 2024-08-09
> > > **Thank you**
> > >
> > > Thank you again for the feedback!

---

### Official Review · Reviewer_QXEW · 2024-07-11

**Soundness:** 3
**Presentation:** 2
**Contribution:** 3
**Rating:** 7
**Confidence:** 4

**Summary:**

This paper proposes a new decoder-only learning scheme for autonomous driving dynamics. Rather than using an encoder-decoder type architecture, the model uses spatial, temporal, and “social” attention between map-agent, time-agent, and agent-agent respectively in a time-autoregressive manner. Further, the work explores the usage of “next patch prediction”, where multiple timesteps are bundled, transformed, and unrolled using an RNN. This allows for easier modelling of long-range dependencies and more efficient processing.

**Strengths:**

**Originality**: To our knowledge, this is the first work in Traffic Simulation utilizing decoder-only autoregressive prediction, so there is a significant degree of novelty.

**Quality** The different “arbitrary” selections, such as the number of neighbors to consider, the top-p to sample and patch size are explicitly ablated, which is nice to see. The modelling of the multiple domains present in autonomous driving seems to be sound.

**Clarity**: The paper is generally structured well and readable.

**Significance**: To our knowledge this is the first fully autoregressive model for autonomous driving simulation. The work is also a clear step up from the previous solutions in the WASAC 2023 challenge.

**Weaknesses:**

**Originality**:  The comparison against previous timeseries-transformers is rather lacking. I would like to see Autoformer and informer mentioned, and, possibly the most relevant here the Space-Time transformer (https://arxiv.org/pdf/2109.12218) which models local interactions within a timestep using graphs (similar to your k-nearest neighbor all2all attention).

**Quality**: Something I would like to see though is a scaling comparison: Transformers tend to perform exponentially better as they scale larger, so seeing something like a 30M model to match the “Trajeglish” work would be nice. It is unclear whether, for instance, the patching is actually necessary to the degree described, or whether this is just an artifact of the model being rather small and shallow. Prior work on the expressivity of transformers, such as https://arxiv.org/abs/2205.11502 has shown that one can map K reasoning steps onto a K-deep model, so it might be that a patching of 10 items is only necessary because the model is too shallow to learn the combination itself. I'm aware that scaling a method might not be possible due to the expense of training a larger model, but without it one cannot be sure of the contributions of patching. I will not see it as a big negative if this is impossible because the performance is good enough to argue for the efficacy of the entire system as a whole, but I'm unsure about the importance of each individual component.

**Clarity**: The notation of the patches is a little hard to parse. I would write
$$S_i^{((\tau-1)\times (l+1))\ :\ (\tau \times l)}$$
 to make the grouping more clear (It took me way too long to mentally group the terms together…). Specifically, it’s hard to visually group the two sides of the “:” together. Figure 1 is slightly confusing: I understand what you want to show, but maybe put boxes to group the timesteps into tokens such that it becomes clear the transformer predicts the next token which is composed of multiple timesteps.
Something I did not quite understand was the tokenization of the map: You say that you sample every 5m and then assign a class to that sample. Does that mean you assign a class to the span from the last sample to the next one (i.e. you assign a class for [0m, 5m], [5m,10m],...) or just at that sample point (i.e. you assign for 0m, 5m, 10m). In the latter case I would assume you can easily jump over crucial information like center lines since those are less than 5m wide.

**Significance**: The small size of BehaviorGPT is nice, but not too interesting for many real world problems: Generally, I would rather have a more accurate 30M model (that e.g. closes the gap to MVTE in acceleration), than a less accurate 3M variant. There are obviously advantages to a model being small, but the runtime difference between a 3M and a 30M parameter model is rarely big enough to make the former worth it (you can run a ResNet 50 with 25M parameters on a raspberry pi in real time…).

**Questions:**

What happens when scaling the model up? Does the need for patching vanish?

What are the results after 20% of training? (particularly interesting because you criticize the data-inefficiency of prior art)

Why are mixing coefficients only predicted once per patch?

Why does training take so much time? (I expect a 3M timeseries model to train a lot more quickly)

Does training time change significantly with larger patches? (due to RNNs needing to be unrolled)

Why only autoregressive across time and not also across e.g. space?

You claim that
> Developed the Next-Patch Prediction scheme to enhance models’ capability of long-range interaction reasoning, leading to more realistic multi-agent simulation over a long horizon;

Can you support this by plotting e.g. accuracy vs. prediction horizon?
As is, it is unclear whether the higher performance is actually due to long-horizon performance or just an improvement in the short term and then matching the long-horizon performance.

More generally: How is the generalisation to longer sequence lengths? Generalisation to longer sequences is a known problem for transformers.

**Limitations:**

All models considered are really small, making it hard to judge whether the individual components of the model are necessary or just an artifact of the expressivity of small models being low.

---

> ### Author Rebuttal · Authors · 2024-08-03
>
> We sincerely appreciate your thoughtful comments. We will revise the statements, notations, and figures according to your suggestions. In the following, we attempt to address your critical concerns by giving more analyses and clarifying the implementation details.
>
> **Q1: Comparisons with timeseries/spatial-temporal Transformers.**
>
> **A1**: Thank you for pointing out these relevant works. We will discuss them in the revised version.
> ***
> **Q2: Scaling experiments.**
>
> **A2**: Experimenting with models as large as Trajeglish is beyond the reach of our computing resources, but we try our best to conduct some scaling experiments under the constraints of limited time and computing budget. We conduct two groups of experiments, including (1) varying the hidden size when fixing the number of decoder layers to be 2 and (2) varying the number of decoder layers when fixing the hidden size to be 128. Although the experimental results show that larger models can achieve better results, the gain in performance also seems to plateau at the scale of 5-7M parameters. we can expect from the trend that continuing to enlarge the model would not bring too significant improvement.
> | Hidden Size | #Param | minADE $\downarrow$ | Realism $\uparrow$ | Collision $\uparrow$  | Offroad $\uparrow$  |
> | :--- | :---: | :---: | :---: | :---: | :---: |
> | 64 | 800K | 1.9637 | 0.7251 | 0.9229 | 0.9056 |
> |128 | 3M | 1.6247 | 0.7349 | 0.9409 | 0.9163 |
> |192 | 7M | 1.4993 | 0.7382 | 0.9439 | 0.9185 |
>
> | #Decoder Layer | #Param | minADE $\downarrow$ | Realism $\uparrow$ | Collision $\uparrow$  | Offroad $\uparrow$  |
> | :--- | :---: | :---: | :---: | :---: | :---: |
> | 1 | 2M | 1.7318 | 0.7319 | 0.9319 | 0.9149 |
> | 2 | 3M | 1.6247 | 0.7349 | 0.9409 | 0.9163 |
> | 3 | 4M | 1.5381 | 0.7387 | 0.9450 | 0.9199 |
> | 4 | 5M | 1.4881 | 0.7396 | 0.9481 | 0.9207 |
> ***
> **Q3: Tokenization of the map.**
>
> **A3**: Each map token spans 5 meters, and a semantic category is assigned to the 5-meter segment. We will clarify this in the revised version.
> ***
> **Q4: Significance of small models.**
>
> **A4**: We understand that today is the era of large models, but we must point out that many industrial applications, such as autonomous driving, are still using small models (some false advertising on large models might somewhat mislead consumers). As also indicated by the Reviewer gNhc, autonomous driving companies need to run simulations at massive scales before real-world deployment, so simulation efficiency determines how fast we can upgrade the autonomous driving system. In industry, pursuing 1% more realistic simulation with 10x larger models may not be a wise strategy. Thus, we believe small models can also make a great impact on real-world applications.
> ***
> **Q5: Does an extremely large model need the patching design?**
>
> **A5**: Since it is impossible for us to train a super-large model, we are unable to reach a conclusion. However, we have shown that the patching design helps small models beat up very large models, demonstrating the value of the patching mechanism--if autonomous driving companies can attain satisfying simulation quality with a small model, there is no reason for them to spend way more money on training and deploying an extremely large model.
> ***
> **Q6: Data efficiency.**
>
> **A6**: We evaluate our models (5M parameters, hidden size = 128, #Decoder layer = 4) trained with different proportions of training data. Our model is able to achieve very decent performance when trained on merely 20% of training data, which is attributed to the high data efficiency of our approach.
> | Training Data | minADE $\downarrow$ | Realism $\uparrow$ | Linear Speed  $\uparrow$  | Linear Acceleration $\uparrow$  | Offroad $\uparrow$  |
> | :--- | :---: | :---: | :---: | :---: | :---: |
> | 20% | 1.4881 | 0.7396 | 0.3633 | 0.3181 | 0.9207 |
> | 50% | 1.4060 | 0.7427 | 0.3637 | 0.3203 | 0.9250 |
> | 100% | 1.3804 | 0.7438 | 0.3655 | 0.3227 | 0.9268 |
> ***
> **Q7: Why are mixing coefficients only predicted once per patch?**
>
> **A7**: Because we desire the mixing coefficients to represent the likelihood of *multi-step* behavior.
> ***
> **Q8: Why does training take so much time?**
>
> **A8**: Besides model size, the training time is also determined by the computational complexity. On the one hand, our model is trained on 9-second sequences at 10 Hz. On the other hand, the agent simulation task is more than a timeseries problem, as most traffic scenarios involve hundreds of agents. Last but not least, the Waymo Open Motion Dataset is one of the largest datasets in autonomous driving, involving 574 driving hours and a 1.4 TB download size.
> ***
> **Q9: Does training time change significantly with larger patches?**
>
> **A9**: We did not notice a significant change in training time when using a patch size of 10, which may be due to the light weight of the RNN head (1 GRU layer with a hidden size of 128).
> ***
> **Q10: Why only model the space dimension autoregressively?**
>
> **A10**: While it is natural to model the time dimension autoregressively, it is difficult to determine the order of agents in the chain. Image autoregressive models (e.g., PixelCNN) also face this problem, where the order of pixels in the chain may significantly affect the performance.
> ***
> **Q11: Long-term vs short-term.**
>
> **A11**: Please note that scenarios are generated autoregressively rather than in one shot. Without good short-term performance, it is unlikely to have good long-term performance owing to compounding errors.
> ***
> **Q12: How about the extrapolation ability to longer sequences?**
>
> **A12**: We tried training a model on 5-second sequences and generating 9-second sequences during inference. The results below demonstrate the extrapolation ability of our approach.
> | Training | Inference | minADE $\downarrow$ | Realism $\uparrow$ | Collision $\uparrow$ | Offroad $\uparrow$ |
> | :--- | :--- | :--- | :--- | :--- | :--- |
> | 9s | 9s | 1.6247 | 0.7349 | 0.9409 | 0.9163 |
> | 5s | 9s | 1.6294 | 0.7333 | 0.9375 | 0.9100 |

---

> > ### Comment · Reviewer_QXEW · 2024-08-08
> >
> > Thank you for your thorough followup!
> >
> > > Scaling experiments / Significance of small models
> >
> > The scaling experiments are very interesting.
> >  The reason I'm so hung up on model size is that up-to a certain size (depending on e.g. the cache size of the GPU) there is only a small impact when increasing the model size significantly, so increasing the model size by a factor X would not affect the runtime by that same factor X. This is especially true during inference where models need much less memory due to the lack of gradients.
> >  I do think that especially the second table still shows improvements as the model's depth is increased (plotting minADE is an almost linear improvement per added layer), but as I mentioned I do not hold the lack of training ressources against this work. It is still interesting that when increasing the depth we could expect further improvements.
> >
> > > Does an extremely large model need the patching design?
> > > if autonomous driving companies can attain satisfying simulation quality with a small model, there is no reason for them to spend way more money on training and deploying an extremely large model.
> >
> > Fair argument.
> >
> > > Data efficiency
> >
> > That's a pretty good result even after 20% of training. I recommend to add this into at least the appendix to support your claim of higher data efficiency (line 86 in your paper).
> >
> > >Why are mixing coefficients only predicted once per patch?
> > > A7: Because we desire the mixing coefficients to represent the likelihood of multi-step behavior
> >
> > Let me check whether I get this right: you effectively treat every chain of RNN calls as one possible "future" and then mix the different paths with your mode weights? I.e. you assume the distribution of paths is multi-modal, while each path in isolation remains unimodal.
> > In that case your modelling makes sense (though I'm not sure how realistic that "unimodal within path" assumption is).
> >
> > > Q8: Why does training take so much time?
> >
> > Oh, I did not consider that the dataset is 1.4TB worth of motion data. In that case it makes sense that training through 100% of the data takes as much time as it does.
> >
> > > Q9: Does training time change significantly with larger patches?
> >
> > Good to know: My concern was that the recursive evaluation of the RNN itself made the training time as long as it took, but if that isn't a problem than that's great.
> >
> > > Q10: Why only model the space dimension autoregressively?
> >
> > Makes sense
> >
> > > Q11: Long-term vs short-term. / How about the extrapolation ability to longer sequences?
> >
> > The model being able to extraplolate to ~2 times its training length is sufficient for me to support your claim in line 87 (maybe also add this into the appendix).
> > It's worth noting that "long range interaction" can mean very different things: In your case "long range" is 90 steps (9s@10Hz) while other transformer papers talk about "long ranges" in the order of 1000 to 16000 steps (e.g. https://arxiv.org/abs/2011.04006). However, I'm also not sure how you can easily clarify this since what people think of "long range" heavily depends on their background.
> >
> >
> > From my end, the authors have adressed all my concerns to the best of their ability and I would increase my score to an accept (7).

---

> > > ### Author Response · Authors · 2024-08-09
> > > **Thanks for your positive feedback!**
> > >
> > > Thank you very much for the positive feedback! We will supplement these important experimental results and continue to improve the quality of our paper as you suggested.
> > >
> > > Below is a bit more discussion regarding your comments:
> > >
> > > **1. Assumption of unimodal within a path.**
> > >
> > > The motivation behind this assumption is that a path represents a high-level intention of agents, with the corresponding mixing coefficient capturing the likelihood of the intention (which we call "intention uncertainty"). Each step of a path is modeled as a unimodal distribution, with the variance of the distribution capturing the "control uncertainty."
> > >
> > > **2. The notion of "long-range."**
> > >
> > > We are aware that the meaning of "long-range" depends on the specific context. For decision making and motion planning in autonomous driving, 9-second sequences are fairly long (imagine yourself as a driver or a pedestrian:  is it trivial to anticipate what will happen on the road 9 seconds later?). Since we have confined our paper to the domain of autonomous driving, we think the notion of long-range interaction would not be that confusing, but we will still clarify the sequence length produced by our model in the revised paper.
> > >
> > > Thank you again for the insightful comment!

---

### Official Review · Reviewer_mLGo · 2024-07-12

**Soundness:** 4
**Presentation:** 3
**Contribution:** 4
**Rating:** 7
**Confidence:** 4

**Summary:**

This work presents BehaviorGPT, a model for trajectory prediction which is decoder only and respects temporal causality by employing a autoregressive sequence model.  They opt for a coarser time resolution of the sequence they call "patching" for reasons of efficiency and larger context, in analogy to word-level instead of byte-level representations for language sequence models. After predicting a coarser time segment ("patch") embedding, they then decode this into the finer time resolution sequence of states also respecting temporal causality with a RNN decoder within the "patch".

The model is decoder only, in the sense that they can process an infinite temporal stream where agent input representation is the same as agent output representation.  For the static map information, this is encoded once and cross-attended to in the rest of the decoder-only model.  The inner architecture is interleaved attention between map, agents, and time dimension.

They report exceptional performance results on the Waymo Open Sim Agents Challenge (WOSAC), which requires the models obey temporal causality to respect a sim agents use case. At the same time, their model is 92% smaller than competitive methods, which is substantially more parameter efficient.

**Strengths:**

This is a significantly novel decoder-only model, with reasonable complexity, with very strong results.

The ideas like "triple attention", QCNet's positional encoding are present in other papers.  Predicting coarser state subsequences is a straightforward idea.  But piecing these ideas together into the full decoder only model results in a significant new model (in the specific area of multi-agent trajectory prediction).

With caveats, the paper is mostly very clearly explained, and will definitely have impact for other practitioners to replicate/extend.

**Weaknesses:**

The biggest doubt I have about the impact of this paper is: is the impact of the paper
A. the decoder only architecture or
B. "patching", aka lower resolution modeling for efficiency reasons

Without patching, the model performs significantly worse than other top methods (compare minADE with patchsize=1 in Table 3 to minADEs in Table 1).  Thus,
 - does decoder-only really matter?
 -  if one added the patching idea to Trajeglish or MVTE or MTR++, would those also perform much better?

My hunch is patching is doing the heavy lifting here, but that is not clear in the story.

If this hunch is true, one story of this paper is "other methods faithfully adhere to the overly-fine 10hz native processing, but BehaviorGPT found a way to bypass that inefficiency"

**Questions:**

Q1. I have a big pet peeve with the terminology "patching", which is why I mostly keep it in quotes throughout this writeup.  To me (as a native english speaker), the notion of a "patch" only makes sense to describe a 2D or 3D region.  Calling a subsequence or segment of a 1D sequence a "patch" is very unintuitive.  In language models and in computer science when referring to 1D sequences in general, more common terms are subsequence, (sub)segment, or chunk.  In my opinion, "chunk" is nice, and gives me intuition that your process is analogous to this: https://chunkviz.up.railway.app/

So, not a question, but please consider this!

Q2. How does this model do on the Waymo Open Motion Dataset's Motion Forecasting benchmark? Many papers in Table 1 also report results there, so I am curious.

---

> ### Author Rebuttal · Authors · 2024-08-02
>
> Thank you for the insightful comments! We fully understand your concerns and hope to explain our motivation as well as some subtle details regarding your doubt.
>
> **Q1: Does decoder-only architecture really matter?**
>
> **A1**: Indeed, it is possible to achieve very good results with an encoder-decoder model, but our paper intends to convey the message that "decoder-only models are sufficient to do very well." In other words, it may not be necessary to design some intricate modules to encode the history and employ some well-designed DETR-like decoders for future prediction as done by the most advanced trajectory prediction models, such as Wayformer, MTR, and QCNet. Given that using a much more concise decoder-only architecture can attain super strong results, there is no reason for us to pursue a more complicated solution due to the principle of Occam's razor. Besides the conciseness, combining decoder-only Transformers with relative spatial-temporal positional embeddings has advantages in engineering, as we can utilize the key-value cache technique to reuse past computations. By contrast, typical encoder-decoder models need to encode the history and run the full decoder at each simulation step, which is quite inefficient.
>
> Moreover, we also found that decoder-only models have very high sample efficiency. Compared with encoder-decoder models where each scenario is treated as one pair of history and future, BehaviorGPT models each time step as the current one and requires each state to model subsequent states’ distribution during training, which is equivalent to constructing training samples from every possible history-future pair of the time series. To illustrate the sample efficiency of BehaviorGPT, we evaluate the models (5M parameters, hidden size = 128, #Decoder layer = 4) trained with different proportions of training data. The table below shows that using merely 20% of the data for training has already enabled the model to surpass many SOTA. The high sample efficiency allows small models to achieve incredible performance, which is also an essential reason for using decoder-only architecture.
> | Training Data | minADE $\downarrow$ | Realism $\uparrow$ | Offroad $\uparrow$  |
> | :--- | :---: | :---: | :---: |
> | 20% | 1.4881 | 0.7396 | 0.9207 |
> | 50% | 1.4060 | 0.7427 | 0.9250 |
> | 100% | 1.3804 | 0.7438 | 0.9268 |
> ***
> **Q2: If one added the patching idea to other models or lowered the replan frequency, would those also perform much better?**
>
> **A2**: We believe the patching idea can be applied to other autoregressive models (e.g., Trajeglish). But one thing we want to point out is that a lot of top methods on the WOSAC are adapted from SOTA motion prediction models (e.g., MTR). These methods do not necessarily comply with the closed-loop requirements of agent simulation and often produce 8-second trajectory points in one shot. Thus, these open-loop methods have already been operating at an extremely low frequency (i.e., 0.125 Hz, which is forbidden). However, our approach outperforms them by a large margin when operating at a moderate replan frequency.
>
> To help readers understand how the replan frequency may affect the simulation results, we varied the replan frequency of the same model (patch size = 10) during inference, which can be achieved by discarding a portion of the predicted states at each simulation step. The test set results are shown as follows. From the table, we can see that increasing the replan frequency from 1 Hz to 2 Hz can improve the overall performance, which may benefit from the enhanced reactivity. This phenomenon demonstrates that the performance gain is not merely due to the lower replan frequency, as the model with a patch size of 10 beats that with a patch size of 5 even harder if using the same replan frequency (i.e., 2 Hz). However, we found that an overly high replan frequency harms the performance, as indicated by the third row of the table. Overall, we conclude that using a larger patch indeed helps long-term reasoning, but a moderate replan frequency is important for temporal stability, which may be neglected by prior works.
> | Patch Size | Replan Frequency | minADE $\downarrow$ | Realism $\uparrow$ | Offroad $\uparrow$  |
> | :--- | :---: | :---: | :---: | :---: |
> | 10 | 1 Hz | 1.5405 | 0.7414 | 0.9308 |
> |10 | 2 Hz | **1.4147** | **0.7473** | **0.9349** |
> | 10 | 5 Hz | 1.5693 | 0.7342 | 0.9089 |
> ***
> **Q3: Is it appropriate to use the terminology "patching"?**
>
> **A3**: Thanks for your comment, and we agree that the notion of a "patch" is more suitable for describing 2D or 3D stuff. In fact, BehaviorGPT uses a multi-agent formulation to simulate all agents' states simultaneously. In this sense, our formulation involves multiple dimensions, including the agent dimension, the time dimension, and the state dimension (3D position + 2D velocity + 1D yaw angle). Thus, we think it appropriate to use the notion of a "patch" in our case.
> ***
> **Q4: What about the motion forecasting results?**
>
> **A4**: As mentioned above, many approaches on the WOSAC leaderboard are adapted from typical marginal motion forecasting models, so they can be tested on the Waymo motion prediction benchmark without any effort. In contrast, BehaviorGPT is a generative model that considers the joint distribution of all agents in the scene, which is misaligned with the objective of the Waymo motion prediction benchmark. Thus, it is more reasonable to compare our approach with other joint multi-agent motion prediction models. In fact, the WOSAC leaderboard also evaluates minADE, the most commonly used metric for motion prediction. We also noted that QCNeXt, a joint multi-agent prediction model that has won 1st place in the CVPR 2023 Argoverse 2 multi-agent motion forecasting challenge, is also on the WOSAC leaderboard. We found that QCNeXt performs much better on minADE (1.08 vs our 1.54), but its closed-loop performance lags far behind most simulation-oriented models. It seems to be a trade-off.

---

### Official Review · Reviewer_b92q · 2024-07-12

**Soundness:** 3
**Presentation:** 3
**Contribution:** 3
**Rating:** 7
**Confidence:** 4

**Summary:**

The paper proposes a decoder only model for traffic simulation. The paper proposes to not predict the next state/token but to predict the next patch a small trajectory segment. The network can select several possible patch generators which are based on RNNs. The resulting architecture achieves strong results with a small transformer based architecture only using about 3M parameters.

**Strengths:**

Proposed approach is interesting and works well. The method combines several ideas that have been shown to be efficient and combines them in a network for traffic simulation. The results of the model given its size are impressive and it would be interesting how it performed if the size was scaled up.

**Weaknesses:**

-The ablation study is not fully clear to me, I can not see a base model, and I also do not know what parameters are used in the final model.

-Most of the ideas in the paper are known in other related fields. Not just predicting the action/next state is known in imitation learning to help with compounding errors. It is also getting used in LMMs [1]. The transformer architecture design is similar to HPTR.

-A lot of details to reproduce the results are not clear (see questions)

[1] Better & Faster Large Language Models via Multi-token Prediction, Gloeckle. et al.

**Questions:**

-Is it correct that you have as many RNNs as modes, with N_mode different parameters?

-How do you select the 32 trajectories for the submission?

-If you keep all modes with p>=0.9 do you not have issues with an excessive amount of trajectories? Or do you just keep the top-1?

-Are there more tricks in the post processing, this is normally important to achieve good results for the sim agent challenge.

-Is it correct that your input tokens are continuous?

-Do you run the patch generation in a receding horizon fashion, where the model would run at 10Hz even though you generate 1s outputs. If not, would it be possible to do so?

-Why is the training so slow? I believed that one of the advantages of a decoder only model compared to more policy learning based methods such as TraffiBot would be training speed. Since there is not autoregressive rollout during the training

-Why do you use a RNN for the patch generation, could a simple MLP not do the same job and be faster?

**Limitations:**

Limitations could be discussed in more details.

---

> ### Author Rebuttal · Authors · 2024-08-05
>
> We sincerely thank you for the valuable feedback. In the following, we answer your questions to resolve some critical concerns.
>
> **Q1: Details of final models and ablation studies.**
>
> **A1**: We list the detailed configurations of each group of experiments as follows, which will be added to the revised version.
> |  Table | Training Data | Evaluation Set | #Param | #Max Agent | #Decoder Layer | Hidden Size | Patch Size | RNN Head | #Neighbor | #Mode | top-p |
> | :--- | :--- | :--- | :--- | :--- | :--- | :--- | :--- | :--- | :--- | :--- | :--- |
> | 1 | 100% | Test | 3M |128 | 2 | 128 | 10 | Autoregressive | 32 | 16 | 1.0 |
> | 3 | 20% | Val | 3M | 128 | 2 | 128 | Ablated | Non-Autogressive | 32 | 16 | 1.0 |
> | 4 | 20% | Val | 3M | 64 | 2 | 128 | 10 | Non-Autogressive | Ablated | 8 | 0.95 |
> | 5 | 20% | Val | 3M | 100 | 2 | 128 | 10 | Non-Autogressive | 32 | Ablated | Ablated |
> ***
> **Q2: Relevant ideas in related fields.**
>
> **A2**: Thank you for mentioning the relevant ideas in related fields, such as the concurrently proposed multi-token prediction in LLMs and the KNN design of HPTR, which will be discussed in the revised version. However, using patch-wise tokens in decoder-only Transformers is a new attempt to our knowledge.
> ***
> **Q3: Implementation Details.**
>
> **A3**: All hidden sizes are set to 128. All attention layers have 8 attention heads with 16 dimensions per head. We train our models for 30 epochs with a batch size of 24 using the AdamW optimizer. The weight decay rate and the dropout rate are both 0.1. Using the cosine annealing scheduler, we decay the learning rate from $5 \times 10^{-4}$ to 0. Our main results are produced by a single model with 2 decoder layers and 3M parameters. Other details that you are interested in are discussed below.
>
> **(1) Is it correct that you have as many RNNs as modes, with N_mode different parameters?**
>
> The parameters of the RNN are shared across all modes.
>
> **(2) Sampling strategy.**
>
> We randomly sample one behavior mode from each agent's next-patch distribution until we complete the 8-second trajectories of all agents. To obtain 32 replicas of rollouts, we repeat this process using different random seeds.
>
> **(3) Post-processing tricks.**
>
> We do not apply any other post-processing tricks.
>
> **(4) Is it correct that your input tokens are continuous?**
>
> Yes, our input and output tokens are all continuous.
>
> **(5) Do you run the patch generation in a receding horizon fashion?**
>
> We try to increase the replan frequency by discarding a portion of the predicted states at each simulation step. The test set results produced by the model with a patch size of 10 are shown as follows. From the table, we can see that increasing the replan frequency from 1 Hz to 2 Hz can improve the overall performance, which may benefit from the enhanced reactivity. However, we found that an overly high replan frequency harms the performance, as indicated by the third row of the table. The results in Table 1 are based on 1-Hz replan frequency.
> | Patch Size | Replan Frequency | minADE $\downarrow$ | Realism $\uparrow$ | Offroad $\uparrow$  |
> | :--- | :---: | :---: | :---: | :---: |
> | 10 | 1 Hz | 1.5405 | 0.7414 | 0.9308 |
> |10 | 2 Hz | **1.4147** | **0.7473** | **0.9349** |
> | 10 | 5 Hz | 1.5693 | 0.7342 | 0.9089 |
>
> **(6) Why is the training so slow?**
>
> Compared with typical encoder-decoder models, the sequence length in our approach is much longer. For example, most motion forecasting models on the Waymo Open Motion Prediction Benchmark utilize 1-second history to predict 8-second future trajectories, while our decoder-only Transformers simultaneously utilize 1-second, 2-second, ..., and 8-second history to predict the next patch during training.
>
> **(7) Why do we use RNNs for the patch generation?**
>
> We hope to comply rigorously with the closed-loop requirements of agent simulation via autoregressive RNNs. We have not tested a simple MLP head yet.
> ***
> **Q4: What are the limitations?**
>
> **A4**: (1) Currently, BehaviorGPT underperforms in kinematics-related performance (e.g., linear/angular speed likelihood), which can be enhanced by incorporating a kinematic model (e.g., bicycle model); (2) BehaviorGPT does not support controlling agent behavior with prompts (e.g., language, goal points). Future work on agent simulation may consider controllable generation; (3) We have not verified whether BehaviorGPT will facilitate the development of motion planning.
> ***
> **Q5: Scaling experiments.**
>
> **A5:** We try to conduct some scaling experiments under the constraints of time and computing budget.
>
> (1) **Hidden size**: We vary the hidden size to obtain models with different parameters. The experiments use 20% of the training data and 2 layers of the Transformer decoder. As depicted below, increasing the hidden size consistently improves the performance.
> | Hidden Size | #Param | minADE $\downarrow$ | Realism $\uparrow$ | Linear Speed  $\uparrow$  | Collision $\uparrow$  | Offroad $\uparrow$  |
> | :---: | :---: | :---: | :---: | :---: | :---: | :---: |
> | 64 | 800K | 1.9637 | 0.7251 | 0.3369 | 0.9229 | 0.9056 |
> |128 | 3M | 1.6247 | 0.7349 | 0.3546 | 0.9409 | 0.9163 |
> |192 | 7M | 1.4993 | 0.7382 | 0.3646 | 0.9439 | 0.9185 |
>
> (2) **Number of decoder layers**: We also try to fix the hidden size as 128 and vary the number of decoder layers, obtaining models by training on 20% of the data. Based on the experimental results below, we can conclude that increasing the depth of the models can benefit the performance.
> | #Decoder Layer | #Param | minADE $\downarrow$ | Realism $\uparrow$ | Linear Speed  $\uparrow$  | Collision $\uparrow$  | Offroad $\uparrow$  |
> | :--- | :---: | :---: | :---: | :---: | :---: | :---: |
> | 1 | 2M | 1.7318 | 0.7319 | 0.3465 | 0.9319 | 0.9149 |
> | 2 | 3M | 1.6247 | 0.7349 | 0.3546 | 0.9409 | 0.9163 |
> | 3 | 4M | 1.5381 | 0.7387 | 0.3570 | 0.9450 | 0.9199 |
> | 4 | 5M | 1.4881 | 0.7396 | 0.3633 | 0.9481 | 0.9207 |

---

> > ### Comment · Reviewer_b92q · 2024-08-13
> >
> > Thanks a lot for addressing my comments, my main concerns are resolved and given that the material of the rebuttal is added I would like to increase my rating to a 7.

---

> > > ### Author Response · Authors · 2024-08-14
> > > **Thank you**
> > >
> > > Thank you again for the constructive suggestions!

---

### Official Review · Reviewer_HMYm · 2024-07-13

**Soundness:** 3
**Presentation:** 3
**Contribution:** 2
**Rating:** 6
**Confidence:** 4

**Summary:**

This paper presents BehaviorGPT, a model for multi-agent traffic simulation. BehaviorGPT's architecture is based on a decoder-only transformer that autoregressively predicts patches of trajectories. The key insight is that predicting patches forces the model to learn longer-horizon reasoning. Then, to predict states from patches, BehaviorGPT uses an RNN that outputs a mixture of Gaussians/Laplacians for each agent's x, y, and heading. BehaviorGPT achieves state-of-the-art performance on the Waymo Open Sim Agents Challenge.

**Strengths:**

- The proposed architecture is simple yet incorporates well-reasoned inductive biases like relative positional encodings and factorized self-attention. Despite its simplicity, BehaviorGPT achieves state-of-the-art performance on the Waymo Open Sim Agents Challenge. Moreover, BehaviorGPT does this with 10x fewer parameters than the previous state-of-the-art, Trajeglish.
- The paper ablates the efficacy of patch-based tokens, convincingly demonstrating that it is critical to having strong performance. To my knowledge, this design choice is novel within multi-agent traffic simulation and its adjacent fields.
- Code will be made available after publication.
- The paper is generally well-written and easy-to-follow.

**Weaknesses:**

- The paper could be strengthened with stronger ablation studies to highlight the design choices that make BehaviorGPT outperform prior work, like Trajeglish. A number of design choices in BehaviorGPT set it apart from Trajeglish, but it is not clear which ones actually contribute to its ability to outperform Trajeglish even with significantly less parameters; e.g., is it the patch-based tokens? is it the factorized attention with relative positional encodings? is it the specific mixture of Gaussians/Laplacians output distribution? Such analysis would give the reader more insight into what design choices are important for multi-agent traffic simulation.
- The paper does not provide sufficient detail to reproduce the model; e.g., basic hyperparameters like the number of transformer layers. That said, this weakness is mitigated by the authors' intention to publish code upon acceptance.

**Questions:**

1. During inference, how do you generate 32 rollouts? Do you randomly sample with a different seed 32 times?
2. What is the significance of not using ground truth state when unrolling the RNN? Do you have experiments to illustrate why it's necessary?
3. Can the authors discuss limitations of their method in a more meaningful manner than "performance can be better"?

**Limitations:**

Limitations are mentioned (in the checklist section only) but limited to "performance could be better". The paper can be improved with a more thorough discussion of limitations; e.g., what are the reasons why BehaviorGPT still underperforms in certain metrics on Table 1?

---

> ### Author Rebuttal · Authors · 2024-08-01
>
> We sincerely appreciate your thoughtful comments and constructive suggestions. In the following, we attempt to address your critical concerns by giving more analyses and clarifying the implementation details.
>
> **Q1: What makes BehaviorGPT outperform SOTA with fewer parameters?**
>
> **A1:** Indeed, the most notable design choices that set BehaviorGPT apart from other autoregressive behavior models lie in three perspectives, including (1) the patch-based tokens, (2) the decoder-only Transformers based on relative spacetime representation, and (3) the use of continuous distributions. Let us understand these design choices better with more in-depth analyses.
>
> (1) **Patching mechanism**: Our paper argues that the Next-Patch Prediction scheme can enhance models’ capability of long-range reasoning. In Table 3, we show that using a larger patch (e.g., increasing the patch size from 5 to 10) can improve the performance. However, some may doubt that the performance gain is merely due to the lower replan frequency (e.g., the 1 Hz for a patch size of 10 compared with the 2 Hz for a patch size of 5) as discovered by Waymo [30], which may enhance the temporal stability of trajectories. To clarify this, we try to increase the replan frequency by discarding a portion of the predicted states at each simulation step. The test set results produced by the model with a patch size of 10 are shown as follows. From the table, we can see that increasing the replan frequency from 1 Hz to 2 Hz can even improve the overall performance, which may benefit from the enhanced reactivity. This phenomenon demonstrates that the performance gain is not merely due to the lower replan frequency, as the model with a patch size of 10 beats that with a patch size of 5 even harder if using the same replan frequency. However, we found that an overly high replan frequency harms the performance, as indicated by the third row of the table. Overall, we conclude that using a larger patch indeed helps long-term reasoning, but a moderate replan frequency is important for temporal stability, which may be neglected by prior works.
> | Patch Size | Replan Frequency | minADE $\downarrow$ | Realism $\uparrow$ | Offroad $\uparrow$  |
> | :--- | :---: | :---: | :---: | :---: |
> | 10 | 1 Hz | 1.5405 | 0.7414 | 0.9308 |
> |10 | 2 Hz | **1.4147** | **0.7473** | **0.9349** |
> | 10 | 5 Hz | 1.5693 | 0.7342 | 0.9089 |
>
> (2) **Decoder-only Transformers with relative spacetime representation**: Decoder-only Transformers with relative spacetime representation can utilize training data more efficiently. Compared with encoder-decoder models where each scenario is treated as one pair of history and future, BehaviorGPT models each time step as the current one and requires each state to model subsequent states’ distribution during training, which is equivalent to constructing training samples from every possible history-future pair of the time series. To illustrate the sample efficiency of BehaviorGPT, we evaluate the models (5M parameters, hidden size = 128, #Decoder layer = 4) trained with different proportions of training data. The table below shows that using merely 20% of the data for training has already enabled the model to surpass many SOTA. The high data efficiency allows small models to achieve incredible performance.
> | Training Data | minADE $\downarrow$ | Realism $\uparrow$ | Offroad $\uparrow$  |
> | :--- | :---: | :---: | :---: |
> | 20% | 1.4881 | 0.7396 | 0.9207 |
> | 50% | 1.4060 | 0.7427 | 0.9250 |
> | 100% | 1.3804 | 0.7438 | 0.9268 |
>
> (3) **Choice of distributions**: Unlike standard autoregressive models, we choose continuous distributions for modeling since we prefer more end-to-end solutions. To model traffic scenarios with categorical distributions, we must first conduct discretization, which involves many essential implementation details (e.g., choosing the proper vocabulary size). Given that the code of Trajeglish is not publicly available, it is inappropriate to recklessly conclude something like "using continuous distributions is better" by comparing our model with a discretized variant that is not well-tuned. However, we indeed found that the choice of distributions is crucial. Prior to using Laplace distributions, we used Gaussian distributions for parameterization and noted that the model could not converge normally (see Figure c in the general response). Moreover, we observed that directly optimizing the full mixture models is better than using the winner-take-all training strategy (Realism score: 0.7396 vs 0.7146). These are the most critical things to make a continuous autoregressive model work well.
> ***
> **Q2: What about the implementation details?**
>
> **A2**: All hidden sizes are 128. All attention layers have 8 attention heads with 16 dimensions per head. We train our models for 30 epochs with a batch size of 24 using the AdamW optimizer. The weight decay rate and the dropout rate are both 0.1. Using the cosine annealing scheduler, we decay the learning rate from $5 \times 10^{-4}$ to 0. Our main results are produced by a single model with 2 decoder layers and 3M parameters. To obtain 32 replicas of rollouts, we randomly sample a behavior mode from each agent's next-patch distribution using different random seeds.
> ***
> **Q3: What is the significance of not using the GT when unrolling the RNN?**
>
> **A3**: We have tried a non-autoregressive GRU that does not rely on the predicted states when unrolling the next states, which can achieve lower minADE (1.5203 vs 1.6554) but performs worse on higher-order kinematic metrics (speed likelihood: 0.3517 vs 0.3544, acceleration likelihood: 0.2630 vs 0.2873).
> ***
> **Q4: What are the limitations?**
>
> **A4**: (1) The kinematic performance can be enhanced by incorporating a kinematic model (e.g., bicycle model); (2) BehaviorGPT does not support controlling agent behavior with prompts (e.g., language, goal points); (3) We have not verified whether BehaviorGPT will facilitate the development of motion planning.

---

> > ### Comment · Reviewer_HMYm · 2024-08-12
> >
> > Thank you for your response. The authors have addressed most of my questions about the paper, particularly as it relates to my question regarding why BehaviorGPT outperforms the state-of-the-art. My only remaining suggestion is that the authors discuss the limitations of this paper in more depth in the camera ready; the current limitations remain surface-level discussions that do not contribute to the reader's understanding of BehaviorGPT's efficacy.
> >
> > Considering this, I would like to maintain my rating and recommend this paper's acceptance.

---

> > > ### Author Response · Authors · 2024-08-12
> > >
> > > Thank you for the feedback. We will add more discussions about this work's limitations as you suggested.

---

### Author Rebuttal · Authors · 2024-08-07

Dear Reviewers,

Please find the qualitative results and the loss curves in the attached PDF file.

Best,

Authors

---

### Decision · Program_Chairs · 2024-09-25

**Decision:**

Accept (poster)

**Comment:**

The paper proposes a decoder-only multi-agent traffic simulation called "BehaviorGPT", which is based on a transformer model (with only 3M parameters) that predicts patches of trajectories (instead of predicting next states/tokens), which are on a coarser time granularity. A recurrent neural network is used as a patch generator that then outputs a mixture of Gaussians/Laplacians for each agent's x, y, and heading, that generates a fine-grained sequence of states.

BehaviorGPT achieves state-of-the-art performance on the Waymo Open Sim Agents Challenge while being considerably smaller than competitive methods.

Decision: All reviewers voted for clear acceptance. I therefore recommend to accept the paper and encourage the authors to use the feedback provided to improve the paper for its final version.